# Intent3D: 3D Object Detection in RGB-D Scans Based on Human Intention

**Weitai Kang**[1]*, **Mengxue Qu**[2] , **Jyoti Kini**[3] , **Yunchao Wei**[2] , **Mubarak Shah**[3] , **Yan Yan**[1]
[1]University of Illinois Chicago, [2]Beijing Jiaotong University, [3]University of Central Florida

## Abstract

In real-life scenarios, humans seek out objects in the 3D world to fulfill their daily needs or intentions. This inspires us to introduce 3D intention grounding, a new task in 3D object detection employing RGB-D, based on human intention, such as "*I want something to support my back.*" Closely related, 3D visual grounding focuses on understanding human reference. To achieve detection based on human intention, it relies on humans to observe the scene, reason out the target that aligns with their intention ("*pillow*" in this case), and finally provide a reference to the AI system, such as "*A pillow on the couch*". Instead, 3D intention grounding challenges AI agents to automatically observe, reason and detect the desired target solely based on human intention. To tackle this challenge, we introduce the new **Intent3D** dataset, consisting of 44,990 intention texts associated with 209 fine-grained classes from 1,042 scenes of the ScanNet [Dai et al., 2017] dataset. We also establish several baselines based on different language-based 3D object detection models on our benchmark. Finally, we propose **IntentNet**, our unique approach, designed to tackle this intention-based detection problem. It focuses on three key aspects: intention understanding, reasoning to identify object candidates, and cascaded adaptive learning that leverages the intrinsic priority logic of different losses for multiple objective optimization.

## 1 Introduction

Performing object detection by following natural language instruction is crucial, as it can enhance the cooperation between humans and AI agents in real-world. Previous studies in Visual Grounding [Kang et al.; 2024a;b; Kazemzadeh et al., 2014; Mao et al., 2016; Plummer et al., 2015; Yu et al., 2016] have focused on localizing objects described by referential language in images. In recent years, research interests have expanded from Visual Grounding in 2D images to 3D indoor scenes, now recognized as 3D Visual Grounding (3D-VG) [Chen et al., 2020; Achlioptas et al., 2020; Wu et al., 2023; Jain et al., 2022; Cai et al., 2022; Luo et al., 2022; Zhao et al., 2021].

In alignment with the sentiment expressed by Wayne Dyer, "*Our intention creates our reality*", humans often identify and locate objects of interest for specific purposes or intentions. A recent demonstration by OpenAI featuring a humanoid robot [OpenAI, 2024] showcases the promising potential of intention-based understanding. As depicted in Fig.1 (left), in the real-world applications, to detect targets based on human intention, 3D-VG requires users to first observe the 3D world, reason about their desired target and provide a reference for the target's category, attributes, or spatial relationship. However, various factors in daily use cases, such as engagement in intensive activities or experiencing visual impairments, may hinder humans from providing referential instructions. Therefore, the ability to automatically infer and detect the desired target based on the user's intentions, as shown in Fig.1 (right), becomes crucial and more intelligent.

Recently, problems akin to reasoning on human instruction have been explored in 2D images, such as intention-oriented object detection [Qu et al., 2024], implicit instructions understanding [Lai et al., 2023], and affordance understanding [Chuang et al., 2018; Luo et al., 2021; Zhai et al., 2022; Lu et al., 2022; Sawatzky et al., 2019]. However, their focus is restricted to a partial view of our reality. Instead, 3D data mirrors the actual world more closely, featuring depth information, complete geometric and appearance characteristics of objects, and a comprehensive spatial context, elements

---

*Code: `https://github.com/WeitaiKang/Intent3D`. Project: `https://weitaikang.github.io/Intent3D-webpage/`.

that are lacking in 2D images, as highlighted in 3D object detection (3D-OD) [Pan et al., 2021; Yang et al., 2018; Misra et al., 2021; Liu et al., 2021b; Jiang et al., 2020; Schult et al., 2022]. Furthermore, understanding 3D spaces plays a crucial role in the advancement of embodied AI [Li et al., 2023b; Gao et al., 2023; Li et al., 2023a], autonomous driving [Huang et al., 2018; Caesar et al., 2020; Cui et al., 2021], and AR/VR applications [Yu et al., 2022; Lim et al., 2022; Liu et al., 2021a]. Therefore, intention-based detection in 3D settings not only better simulates and addresses real-world complexities but also holds greater promise for future advancements.

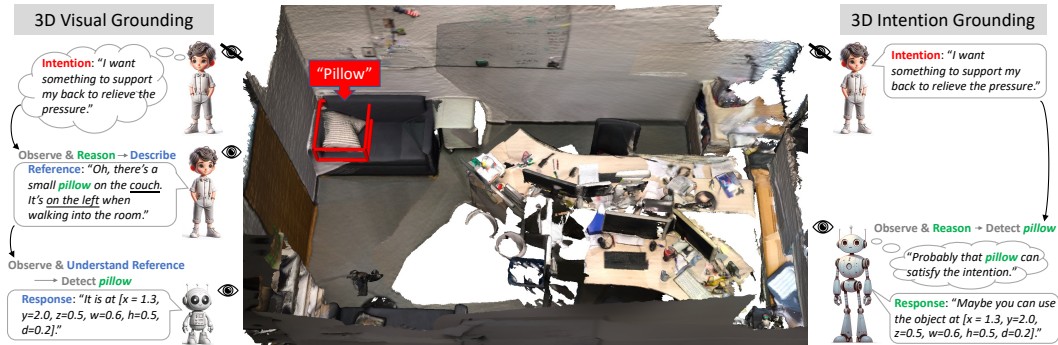

Figure 1: We introduce 3D intention grounding (right), a new task for detecting the object of interest using a 3D bounding box in a 3D scene, guided by human intention expressed in text (e.g., "*I want something to support my back to relieve the pressure*"). In contrast, the existing 3D visual grounding (left) relies on human reasoning and references for detection. The illustration clearly distinguishes that observation and reasoning are manually executed by human (left) and automated by AI (right).

In this paper, we introduce the 3D Intention Grounding (3D-IG) task, which focuses on using human intention cues to automate reasoning and detection of desired objects in real-world 3D scenes. Firstly, we devise a data generation strategy to collect our 3D-IG dataset, named **Intent3D**. Human intentions are diverse. Therefore, it limits generalization if we use predefined formats for human annotators to label the intention sentences. Moreover, as highlighted in [Yang et al., 2024; Abdelreheem et al., 2024; Ding et al., 2022; Tan et al., 2024], crowd-sourced annotators often lack reliability, while professional annotators are expensive and not scalable for large-scale development. Instead, human quality checks [Yang et al., 2024] show that LLMs (GPT-4 [OpenAI]) can serve as human-level annotators, achieving a low error rate. Given these two reasons, we adopt ChatGPT, enriched with diverse daily life knowledge, to generate intentions for each object. Our **Intent3D** dataset comprises 44,990 intention texts representing 209 fine-grained classes of objects from 1,042 ScanNet [Dai et al., 2017] point cloud scenes. Note that 3D-IG is a multi-instance detection problem. For example, given "*I want to present slides to my coworkers,*" we can infer multiple instances of "*Monitor*" in Fig. 1 (right part), making 3D-IG even more challenging. Secondly, to fully gauge our current research capability in solving this problem, we construct several baselines using three main language-based 3D-OD techniques for our benchmark. This involves evaluating our dataset using the following models: expert models specifically designed for 3D visual grounding, an existing foundation model formulated for generic 3D understanding tasks, and a Large Language Model (LLM)-based model. We evaluate these baselines with multiple settings, namely training from scratch, fine-tuning, and zero-shot. Lastly, we propose **IntentNet**, a novel method to tackle the 3D-IG problem. To achieve a comprehensive understanding of intention texts, we introduce the Verb-Object Alignment technique, which focuses on initially recognizing and aligning with the verb and subsequently aligning with the corresponding object features. We further propose the Candidate Box Matching component to explicitly identify relevant object candidates in the sparse 3D point cloud. To concurrently optimize the intention understanding, candidate box matching, and detection objectives, we incorporate a novel Cascaded Adaptive Learning mechanism to adaptively scale up different losses based on their intrinsic priority logic.

In summary, our contributions are threefold: (*i*) We introduce a new task, 3D Intention Grounding (3D-IG), which takes a 3D scene and a free-form text describing the human intention as input to directly detect the target of interest. To construct a benchmark for the 3D-IG task, we compile a new dataset called **Intent3D**, comprising 44,990 intention texts covering 209 fine-grained classes of objects from 1,042 3D scenes. (*ii*) We construct comprehensive baselines for our Intent3D, encompassing three main classes of language-based 3D-OD methods: expert model, foundation model,

and LLM-based model. (*iii*) We propose a novel method, **IntentNet**, for intention-based detection, achieving state-of-the-art performance on the Intent3D benchmark.

## 2 RELATED WORK

**Referential Language Grounding** Visual Grounding (VG) [Kang et al.; 2024a;b; Kazemzadeh et al., 2014; Mao et al., 2016; Plummer et al., 2015; Yu et al., 2016] aims to detect the object by a bounding box, given a 2D image and a human referential instruction pertaining to the object of interest. Utilizing point cloud scans from ScanNet [Dai et al., 2017], ScanRefer [Chen et al., 2020] recently extended this task to 3D Visual Grounding (3D-VG), aiming to predict a 3D bounding box for the target object given a 3D point cloud of the indoor environment and a human referential language instruction. Similarly, ReferIt3D [Achlioptas et al., 2020] proposes the 3D-VG task also based on ScanNet [Dai et al., 2017] but with a simpler configuration, where it supplies ground truth boxes for all candidate objects in the scene and only needs method to choose the correct bounding box. Nevertheless, existing 3D-VG benchmarks predominantly evaluate the capability of methods in interpreting human referential instructions, typically encompassing the category, attributes, or spatial relationship of the object. The exploration of directly inferring and detecting the desired target based on human intention instructions in the 3D domain remains an uncharted area.

**Affordance Understanding** Derived from the concept in [Gibson, 2014], affordance detection aims to localize an object or a part of object in 2D [Chuang et al., 2018; Luo et al., 2021; Lu et al., 2022; Sawatzky et al., 2019] or 3D [Zhai et al., 2022; Deng et al., 2021] focusing on feasible/affordable actions that can be executed while engaging with the object of interest. However, the affordable action still cannot convey human intention; for example, the action "*grasp a bottle*" does not necessarily equate to the underlying intention "*I want something to drink*". Furthermore, the dependency on phrase expressions [Lu et al., 2022], closed-set classifications [Chuang et al., 2018; Luo et al., 2021; Zhai et al., 2022; Sawatzky et al., 2019], and object-centered approaches [Deng et al., 2021] restricts the ability of affordance detection to interpret free-form intention and infer 3D scenes.

**Complex Instruction Understanding** To explore more complex instruction understanding, RIO [Qu et al., 2024] introduces a dataset for intention-oriented object detection based on MSCOCO [Lin et al., 2014] 2D images. Moreover, previous studies have also utilized large language models (LLMs) to address instruction reasoning in 2D images. LISA [Lai et al., 2023] adopts LLM to understand complex and implicit query texts. Ferret [You et al., 2023] reasons about multiple targets involved in instructions and introduces a dataset enriched with spatial knowledge. However, reasoning in 3D scene based on human intentions, remains underexplored.

**Language-based 3D Object Detection Methods** Currently, the 3D-VG community has developed various expert (specialist) models [Chen et al., 2020; Achlioptas et al., 2020; Wu et al., 2023; Jain et al., 2022; Cai et al., 2022; Luo et al., 2022; Zhao et al., 2021], trained from scratch to comprehend the free-form language and sparse point clouds for detection task. State-of-the-art expert models like BUTD-DETR [Jain et al., 2022] and EDA [Wu et al., 2023] inherit the merits from MDETR [Kamath et al., 2021], focusing on understanding the target's category or attributes expressed in the language. 3D-VisTA [Zhu et al., 2023], on the other hand, introduces a transformer-based foundation model to jointly train across multiple 3D tasks. Furthermore, inspired by advancements made in LLMs [Liu et al., 2023; Shang et al., 2024; Yuan et al., 2024], researchers have also explored the utilization of LLMs in 3D domain. Models such as PointLLM [Xu et al., 2023] and Point-Bind [Guo et al., 2023] primarily focus on object point cloud scans, while 3D-LLM [Hong et al., 2024], Chat-3D [Wang et al., 2023], and Chat-3D v2[Huang et al., 2023] are designed for 3D environment scans. We comprehensively evaluate these methods on our Intent3D benchmark. Unlike previous approaches, our proposed IntentNet explicitly models intention-based multimodal reasoning.

## 3 3D INTENTION GROUNDING

Here, we begin by defining the task of 3D Intention Grounding (3D-IG) in Section 3.1. Next, in Section 3.2, we delve into the construction of the Intent3D dataset, which serves as the benchmark for 3D-IG. Lastly, we provide comprehensive statistics for our dataset in Section 3.3.

### 3.1 TASK DEFINITION

3D Intention Grounding aims to detect 3D objects within a 3D scene to match a given human intention. It takes as input a 3D scene (e.g. "*living room*"), represented by a point cloud, and free-form

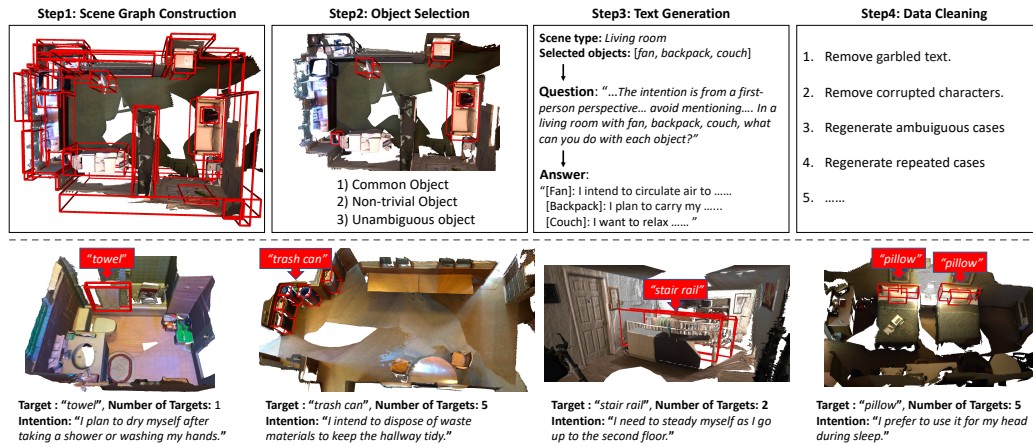

Figure 2: (Upper row) Flowchart for dataset construction. After constructing the scene graph, we select objects by three criteria: Common Object, Non-trivial Object, Unambiguous Object. We use ChatGPT to generate intention texts given the prompt we designed. Finally, we manually clean the data. (Lower row) Examples in our dataset for different number of targets and length of texts.

text (e.g. "*I need a place to sit down and relax*") describing a human intention. The goal is to predict all the 3D bounding boxes for all instances of the target object (i.e. detect all the "*chairs*").

## 3.2 INTENT3D DATASET

The pipeline for our dataset construction is shown in Fig. 2 (upper row).

**Step 1: Scene Graph Construction** First of all, given a 3D point cloud of a scene, we construct a scene graph as a json file to organize all the annotations or information, including the type of the scene, the category at different levels of granularity for each object within the scene, the number of instances sharing the same category, and the 3D bounding box (x, y, z, w, l, h) for each object.

**Step 2: Object Selection** We select the objects in each scene based on the following three criteria. 1) Common Objects: We focus on common indoor objects with diverse human intentions, excluding structural elements like "*wall*" and "*ceiling*", and opting for objects occurring in at least four distinct scenes. 2) Non-trivial Objects: In each scene, we select objects with fewer than six instances in their fine-grained categories, thus excluding overly apparent objects and enhancing the challenge of our grounding task. For example, in a conference room, we select the "*table*" with only one instance rather than the "*chair*" with ten instances. 3) Unambiguous objects: Addressing ambiguity is crucial in generating human intention text, as objects with different fine-grained classes can fulfill the same human intention due to shared higher-level categories (e.g., "*floor lamp*" and "*desk lamp*" both classified under "*lamp*"), synonyms (e.g., "*trash bin*" vs. "*garbage bin*"), and variations between singular and plural terms (e.g., "*bookshelf*" vs. "*bookshelves*"). Furthermore, ambiguity also arises when objects belong to different higher-level categories but can fulfill the same human intention (e.g., both "*TV*" and "*Monitor*" can satisfy the intention "*I want to display my chart to my co-workers*"). To mitigate this, we manually identify ambiguous situations and filter out objects that serve similar human intentions.

**Step 3: Text Generation** We leverage ChatGPT (GPT-4) to generate intention texts in a question-answering manner. The question has two parts: one for constraints (Prompt-1) and the other for scene context (Prompt-2). Specifically, the first part is designed to emulate human expressions of their intentions and prevent the disclosure of referential details, such as the object's category or location, in the response. This is done to challenge AI agents to engage in reasoning independently:

Prompt-1: "*You are a helpful assistant in providing human intention towards each object. The intention is from a first-person perspective, in the format of 'I want / need / intend / ... to ...'. The intention must avoid mentioning synonyms, categories, locations, or attributes of the object.*"

In the second part, we supply ChatGPT with relevant information about the scene:

Prompt-2: "*In a <scene> with <objects>, what can you do with each object?*"

where $<scene>$ refers to the type of scene and $<objects>$ implies a list of all object classes present in the scene. On average, we generate six intention texts per object. Fig. 2 (lower row) displays some samples we generated with varying lengths and different numbers of targets.

**Step 4: Data Cleaning**  Due to occasional nonsensical responses from ChatGPT, we manually filter out gibberish from the generated text. Furthermore, we discard responses that fail to generate. In cases of ambiguity, such as "*keyboard*" in our dataset, where it should refer to a computer keyboard rather than a musical instrument, we manually recreate the sentences. Additionally, in samples with repeated intention sentences, we manually regenerate them to ensure diversity. In our Appendix A, we provide more details (A.1) of failed responses from ChatGPT and the usage of API, and we further discuss ambiguous problem (A.4) and human-likeness test (A.5) of our intention sentences.

## 3.3 DATASET STATISTICS

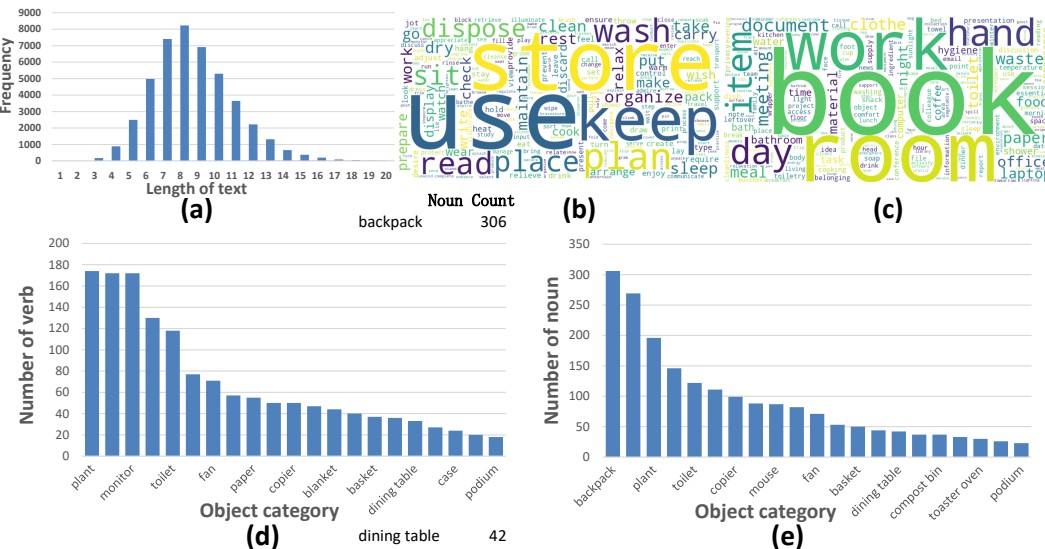

Figure 3: (a) The distribution of the text lengths of all intentions; (b) Word cloud of verbs used in all intention texts; (c) Word cloud of nouns used in all intention texts; (d) Number of different verbs used for each fine-grained class; (e) Number of different nouns used for each fine-grained class.

We have generated a total of 44,990 intention texts for 209 fine-grained classes from 1,042 ScanNet scenes. The dataset includes 63,451 instances and 7,524 distinct scene-object pairs. On average, each scene contains 61 instances and 43 texts, with each scene-object pair receiving 6 intention texts. Given that verbs and nouns play pivotal roles in expressing intentions, we conduct an in-depth analysis of these parts of speech. Specifically, we identify a total of 1,568 different verbs and 2,894 different nouns. The greater variety of nouns compared to the number of object categories suggests that the language of intention is diverse and incorporates a wide-ranging vocabulary from daily life. On average, the intention texts associated with each object category use 58 different types of verbs and 77 different types of nouns. For a more granular understanding, we offer detailed statistics in Fig. 3. Specifically, (a) shows the distribution of text lengths; (b) presents a word cloud of the verbs of all intention texts, excluding the template-derived verbs "*want*", "*need*", "*aim*", and "*intend*"; (c) shows a word cloud of the nouns, omitting the template-derived pronoun "*I*"; (d) and (e) present the number of distinct verbs and nouns used for each fine-grained class in descending order, with every tenth data point displayed for clarity. The dataset is split into train, val, and test sets, containing 35850, 2285, and 6855 samples, respectively, each with disjoint scenes. Our train set comes from ScanNet's train split, while the val and test sets are derived from its val split.

## 4 OUR METHOD

We introduce the structure of IntentNet in Section 4.1. Given that 3D-IG poses a challenge in multimodal reasoning, demanding concurrent engagement in 3D perception, intention comprehension, and joint supervision of multiple objectives, we incorporate Candidate Box Matching for 3D under-

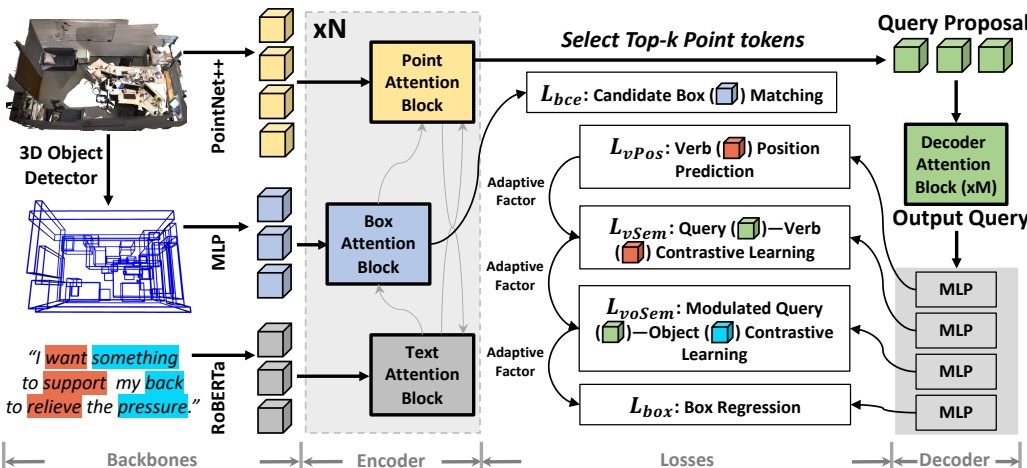

Figure 4: IntentNet: (**Backbones**) PointNet++ is used to extract point features, MLP encodes the boxes predicted by a 3D object detector, and RoBERTa encodes the text input. (**Encoder**) Attention-based blocks are used for multimodal fusion, enhancing box features through integration with text features. (**Decoder**) Point features with top-k confidence are selected as proposed queries and then updated by attention-based blocks. Several MLPs are used to linearly project the queries for subsequent loss calculations. (**Losses**) The model learns to match the candidate boxes with target objects using $L_{bce}$. Queries are trained to identify verbs ($L_{vPos}$), align with verbs ($L_{vSem}$), and align with objects ($L_{voSem}$). Cascaded adaptive factors, which hierarchically weigh each loss based on its dependency on previous losses, are used for optimization. The colored boxes in the loss blocks represent different feature tokens.

standing (Section 4.2), Verb-Object Alignment for text insight (Section 4.3), and Cascaded Adaptive Learning for joint supervision (Section 4.4). We elaborate on our pipeline below.

## 4.1 MULTIMODAL FEATURE EXTRACTION

**Backbones** As shown in Fig. 4 (Backbones), we use the pretrained PointNet++ [Qi et al., 2017] and RoBERTa [Liu et al., 2019] as point cloud and language backbones to extract point features $P \in \mathbb{R}^{p \times c}$ and text features $T \in \mathbb{R}^{l \times c}$ respectively. We also use a pretrained detector, GroupFree [Liu et al., 2021b], to detect 3D candidate boxes as additional visual reference. These candidate bounding boxes are subsequently encoded as box features $B \in \mathbb{R}^{b \times c}$ by MLP module.

**Encoder** As shown in Fig. 4 (Encoder), the point, box, and text features undergo N = 3 layers of attention-based blocks for fusion. In the Point Attention Block, point features perform self-attention and then cross-attention with text and box features. In the Box Attention Block, box features are enhanced by cross-attention with text features for the following matching objective. In the Text Attention Block, text features perform self-attention and then cross-attention with point features.

**Decoder** As shown in Fig. 4 (Decoder), similar to [Jain et al., 2022; Wu et al., 2023], point features are projected to predict whether they are within target boxes. We select the top 50 point features, linearly project them as queries, and update them across M = 6 Decoder Attention Blocks, where queries are involved in self-attention, cross-attention with text features, cross-attention with box features and cross-attention with point features. Finally, the queries are fed into four MLPs to compute the alignment with verb and object text tokens, and predict the boxes. The box regression loss $L_{box}$ is the combination of L1 loss and 3D IoU loss, same as [Jain et al., 2022; Wu et al., 2023].

## 4.2 CANDIDATE BOX MATCHING

Unlike 3D-VG which explicitly specifies target object in its referential language, 3D-IG requires models to automatically infer the desired object in the scene. Therefore, to tackle this reasoning challenge, it is necessary to explicitly identify candidate boxes in close proximity to the target for improved 3D perception associated with the intention. In Fig. 4 (Losses) **Candidate Box Matching**,

matched candidate boxes are computed by minimizing the binary cross entropy (BCE):

$$Confidence = \text{Sigmoid}(\text{MLP}(B)), \ L_{bce} = \text{BCE}(Confidence, GT), \tag{1}$$

where MLP followed by Sigmoid projects box tokens $B$ into confidence scores. $L_{bce}$ is the BCE. $GT$ is the label, which is set to 1 for boxes exceeding 0.25 IoU threshold and 0 otherwise.

## 4.3 Verb-Object Alignment

Unlike referential language, where a single noun primarily denotes the target object, intention language demands that the model concurrently comprehends all verb-object pairs to fully grasp the intended behavior (see Fig. 4 (lower left)), such as "*want*"–"*something*", "*support*"–"*back*", and "*relieve*"–"*pressure*". Evidence of this can be seen in the statistics 3.3, where each object is described by a variety of verbs and nouns. Therefore, as seen in Fig. 4 (Losses), we propose a three-step approach to enhance intention understanding. ❶ The queries develop an awareness of the verb tokens within the intention text through **Verb Position Prediction**. Specifically, we use part-of-speech tagging and dependency parsing from spaCy [Honnibal et al., 2020] to extract the position of the verbs and their corresponding objects (nouns). For example, "*010100*" indicates a 6-token sentence with the verbs appearing at the second and the fourth position. To recognize the verbs within the intention text, the queries are fed into MLP to predict the distribution of verbs, while the label is obtained by applying softmax to the verbs' position:

$$V_{dist} = \text{Softmax}(V_{pos}), \ L_{vPos} = \text{CE}(\text{MLP}(Q), V_{dist}), \tag{2}$$

where $V_{pos} \in \mathbb{R}^l$ is verbs' position, $Q$ is query, $L_{vPos}$ is Cross Entropy loss. ❷ The queries align with the semantics of the verb tokens using **Query-Verb Contrastive Learning**:

$$L_{vSem} = \sum_{n=1}^{k} -\log\left(\frac{\exp(\frac{1}{|\mathcal{T}_l|}\sum_{t\in\mathcal{T}_l}(q_n^T t))}{\sum_{t\in\mathcal{T}}\exp(q_n^T t)}\right) + \sum_{m=1}^{lv} -\log\left(\frac{\exp(\frac{1}{|\mathcal{Q}_l|}\sum_{q\in\mathcal{Q}_l}(t_m^T q))}{\sum_{q\in\mathcal{Q}}\exp(t_m^T q)}\right), \tag{3}$$

where $\mathcal{T}$ and $\mathcal{Q}$ are the set of all the text tokens and queries, which are linearly projected to the same dimension. $\mathcal{T}_l$ and $\mathcal{Q}_l$ are their subsets for the linked verb tokens and queries. The matched queries from bipartite matching [Carion et al., 2020] are linked with the verb tokens. We append "*not mentioned*" at the end of the sentence, and link unmatched queries with the "*not mentioned*" token. $k$ and $lv$ are the number of queries and verb tokens. ❸ To further comprehend the verb-object relationship, the queries need to infer the object token using the provided verb token via **Modulated Query-Object Contrastive Learning**. Specifically, both the queries and verb tokens undergo linear projection. For each verb-object combination in the sentence, the queries are modulated by element-wise multiplication with the verb token. Then, we employ the same contrastive learning in Eq. 3 to align the modulated queries with the corresponding object token, denotes as $L_{voSem}$. In inference, we get a confidence score based on $Softmax(Q^T T_{verb})$, where $Q$ is query and $T_{verb}$ is verb tokens. We omit temperature and normalization in the similarity calculation above for ease of reading.

## 4.4 Cascaded Adaptive Learning

Since we deal with multiple losses, optimizing all the objectives simultaneously becomes challenging. However, there is an inherent chain of logic among these objectives. Intuitively, the loss $L_{vPos}$ should be addressed before $L_{vSem}$, as the queries should first identify the verbs before they can understand their semantics. Similarly, $L_{vSem}$ should precede $L_{voSem}$, as understanding the semantics of the verbs is necessary before inferring the object token of that verb token. Finally, $L_{voSem}$ is a prerequisite for $L_{box}$, as understanding the verb-object combination is essential for intention understanding before intention-related detection. Inspired by Focal Loss [Lin et al., 2017], adaptively adjusting the loss based on difficulty can effectively optimize the learning process and make the best use of training data. Therefore, we use the aforementioned logical chain to create the Cascaded Adaptive Learning scheme, which turns a higher-priority loss into a factor that helps adjust the next lower-priority loss. Specifically, for a given loss, say $Loss_A$, we compute the adaptive factor using $f(x) = x.sigmoid() + 0.5$ (0.5 makes sure the factor is larger than 1). Then, we modulate the next loss, $Loss_B$, by multiplying it with this factor: $Loss_B = Loss_B * f(Loss_A)$. As shown in Fig. 4 (Losses), we cascade this process, i.e. $L_{vPos} \rightarrow L_{vSem} \rightarrow L_{voSem} \rightarrow L_{box}$.

Table 1: 3D intention grounding results on Intent3D's val set. [0] indicates the zero-shot results. The best results are in **bold**, and the second best results are underlined.

| Method | Detector | Top1-Acc@0.25 | Top1-Acc@0.5 | AP@0.25 | AP@0.5 |
|---|---|---|---|---|---|
| BUTD-DETR [Jain et al., 2022] | GroupFree [Liu et al., 2021b] | 47.12 | 24.56 | 31.05 | 13.05 |
| EDA [Wu et al., 2023] | GroupFree [Liu et al., 2021b] | 43.11 | 18.91 | 14.02 | 5.00 |
| 3D-VisTA [Zhu et al., 2023] | Mask3D [Schult et al., 2022] | 42.76 | 30.37 | 36.1 | 19.93 |
| Chat-3D-v2[0] [Huang et al., 2023] | PointGroup [Jiang et al., 2020] | 5.86 | 5.24 | 0.15 | 0.13 |
| Chat-3D-v2 [Huang et al., 2023] | PointGroup [Jiang et al., 2020] | 36.71 | 32.78 | 3.23 | 2.58 |
| IntentNet (Ours) | GroupFree [Liu et al., 2021b] | **58.34** | **40.83** | **41.90** | **25.36** |

Table 2: 3D intention grounding results on Intent3D's test set. [0] indicates the zero-shot results. The best results are in **bold**, and the second best results are underlined.

| Method | Detector | Top1-Acc@0.25 | Top1-Acc@0.5 | AP@0.25 | AP@0.5 |
|---|---|---|---|---|---|
| BUTD-DETR [Jain et al., 2022] | GroupFree [Liu et al., 2021b] | 47.86 | 25.74 | 31.41 | 13.46 |
| EDA [Wu et al., 2023] | GroupFree [Liu et al., 2021b] | 44.00 | 19.62 | 14.56 | 5.18 |
| 3D-VisTA [Zhu et al., 2023] | Mask3D [Schult et al., 2022] | 43.88 | 31.44 | 37.29 | 22.00 |
| Chat-3D-v2[0] [Huang et al., 2023] | PointGroup [Jiang et al., 2020] | 5.63 | 4.93 | 0.14 | 0.11 |
| Chat-3D-v2 [Huang et al., 2023] | PointGroup [Jiang et al., 2020] | 33.46 | 29.32 | 2.67 | 2.1 |
| IntentNet (Ours) | GroupFree [Liu et al., 2021b] | **58.92** | **42.28** | **44.01** | **27.60** |

## 5 EXPERIMENTS

### 5.1 BASELINES

We construct baselines by adopting three main types of language-related 3D object detection methods for the 3D-IG challenge: two expert models, one foundation model, and one LLM-based model.

**Expert model** We adopt two existing 3D-VG expert methods, BUTD-DETR [Jain et al., 2022] and EDA [Wu et al., 2023]. These models use PointNet++ [Qi et al., 2017] and RoBERTa [Liu et al., 2019] as point cloud and language backbones. Additionally, they incorporate box candidates predicted from a pretrained detector [Liu et al., 2021b] as additional visual reference and use transformer layers for multimodal fusion. Limited point features are then chosen for the decoding and prediction stages. We train these models from scratch.

**Foundation model** We fine-tune the SOTA foundation model, 3D-VisTA [Zhu et al., 2023], on our benchmark. It uses a pretrained 3D segmenter [Schult et al., 2022] to segment candidate objects and PointNet++ to extract each object's point clouds. The text branch uses four layers of BERT [Devlin et al., 2018]. A stack of transformer layers is used for multimodal fusion. The model undergoes pre-training across various 3D tasks with unsupervised losses. For detection, it is fine-tuned to identify the best match instances and it uses boxes from segmentation as detection results.

**LLM-based model** Chat-3D v2 [Huang et al., 2023] uses a 3D segmenter [Jiang et al., 2020] to identify object candidates and encode them with a pretrained 3D encoder [Zhou et al., 2023]. It then projects object features into the LLM space and fine-tunes Vicuna-7B [Chiang et al., 2023], incorporating attribute, relation, and object ID learning. Object ID strings are extracted from LLM responses and the corresponding boxes, derived from segmentation, serve as detection outputs. We follow their instructions in the code, thereby formatting all instance annotations into question-answering scripts for training. During evaluation, we calculate the confidence score for each ID string by applying Softmax to the logits of each token prediction and selecting the highest value as the token's score. The final confidence score for each object ID is the product of these token scores.

### 5.2 IMPLEMENTATION DETAILS

For baselines, we follow their configuration on ScanRefer [Chen et al., 2020] by their official codes. However, for BUTD-DETR [Jain et al., 2022], we adopt the implementation from EDA [Wu et al., 2023] since Intent3D does not provide text span labeling, which has been further discussed in [Wu et al., 2023]. We train these two expert models from scratch. For 3D-VisTA [Zhu et al., 2023], we adopt their pretrained checkpoint and fine-tune it. For Chat-3D-v2 [Huang et al., 2023], we also use their pretrained checkpoint and conduct both zero-shot evaluation and fine-tuning based on official settings. Our IntentNet is trained from scratch for 90 epochs with a batch size of 24. The learning rate is 0.001 for PointNet++ and 0.0001 for the rest of the network, which decays by 0.1 at the 65th epoch. The RoBERTa is frozen. The number of point tokens is 1024, and the maximum length for text tokens is set to 256. The hidden dimension used is 288. All the methods are evaluated on the val and test set of Intent3D using their best checkpoint based on the AP@0.5 from the val set.

Table 3: Ablation studies on Intent3D's val set. "Verb" indicates the alignment with verb tokens in the sentence, which is learned by $L_{vPos}$ and $L_{vSem}$. "Verb2Obj" indicates the alignment with object token in the sentence when the queries are modulated by the verb. "MatchBox" indicates the Candidate Box Matching, "Adapt" indicates the Cascaded Adaptive Learning.

| ID | Verb | Verb2Obj | MatchBox | Adapt | Top1-Acc@0.25 | Top1-Acc@0.5 |
|----|------|----------|----------|-------|---------------|--------------|
| (a) |      | ✓        | ✓        | ✓     | 53.09         | 34.62        |
| (b) | ✓    |          | ✓        | ✓     | 57.87         | 39.42        |
| (c) | ✓    | ✓        |          | ✓     | 56.25         | 38.37        |
| (d) | ✓    | ✓        | ✓        |       | 57.39         | 36.93        |
| (e) | ✓    | ✓        | ✓        | ✓     | 58.34         | 40.83        |

## 5.3 EVALUATION METRICS

We use Top1-Accuracy and Average Precision for evaluation. Top1-Accuracy measures the accuracy of the model's highest-confidence prediction, while Average Precision evaluates precision across varying confidence thresholds. We compute the Intersection over Union (IoU) between predicted and ground truth boxes with thresholds of 0.25 and 0.5 to indicate correctness.

## 5.4 QUANTITATIVE COMPARISONS

Tab. 1 and Tab. 2 show quantitative comparisons on val and test set of Intent3D.

**Results of our IntentNet** Due to the explicit modeling of intention language comprehension and reasoning on box candidates with cascaded optimization, our IntentNet significantly outperforms all previous methods. Compared to the second-best approach on the validation set, it achieves improvements of 11.22% and 8.05% in Top1-Acc@0.25 and Top1-Acc@0.5, respectively. Additionally, it improves AP@0.25 and AP@0.5 by 9.12% and 5.43%, respectively. Similarly, on the test set, we obtain 11.06%, 10.84% improvement in Top1-Acc@0.25 and Top1-Acc@0.5, respectively; and 6.72%, 5.6% improvement in AP@0.25 and AP@0.5, respectively.

**Results of expert models** The expert models mostly focus on aligning with nouns in the sentence, since they were originally built for targeting referential language. However, in our intention language, the noun is not the target but only one aspect of an object associated with a certain verb. Therefore, they perform inferiorly compared with our IntentNet, where we explicitly comprehend all the verb-object relations. Notably, a better 3D-VG model, EDA, performs worse than its baseline, BUTD-DETR, since EDA additionally focuses the alignment with the auxiliary object related to the noun in the sentence, which is not the target in our case and thus misleads the model.

**Results of foundation model** The foundation model, 3D-VisTA, performs better than the expert models on most metrics, which is attributed to its widely multimodal alignment in pretraining. However, it still falls short of our IntentNet because it fully relies on detection outputs from the detector [Schult et al., 2022], which are not perfect for evaluation. Instead, we only take the predictions from the pretrained detector as visual reference and provide reasoning on those box candidates. Therefore, even if we are using a less powerful detector [Liu et al., 2021b], we still outperform them.

**Results of LLM-based model** The LLM-based model, Chat-3D-v2, performs the worst in both zero-shot and finetuning on most metrics, which is expected given that current LLM-based models perform poorly on 3D-VG [Hong et al., 2024] and 3D-IG is even more challenging. The main bottleneck might still be the lack of training data for aligning 3D visual features into LLM space [Hong et al., 2024; Huang et al., 2023]. Also, due to hallucination problems in the LLM, Chat-3D-v2 performs significantly worse on AP metrics, which are more sensitive to the false positive predictions. Its decent performance on Top1-Acc is mainly due to the use of a pretrained detector [Jiang et al., 2020] and the powerful LLM to reason out the correct category of the target.

An additional comparison with GPT-4 is elaborated in Appendix A.6.

## 5.5 ABLATION STUDIES

Ablation studies are conducted on the val set to evaluate the effectiveness of each component in our IntentNet. As shown in Tab. 3, we apply controlled variables, removing one component at a time for comparison with the complete version (e). We observe that every component is essential for optimal performance, since the exclusion of any component results in a notable decline in performance. Specifically, the comparison between (a) and (e) shows that the alignment with verb tokens in the

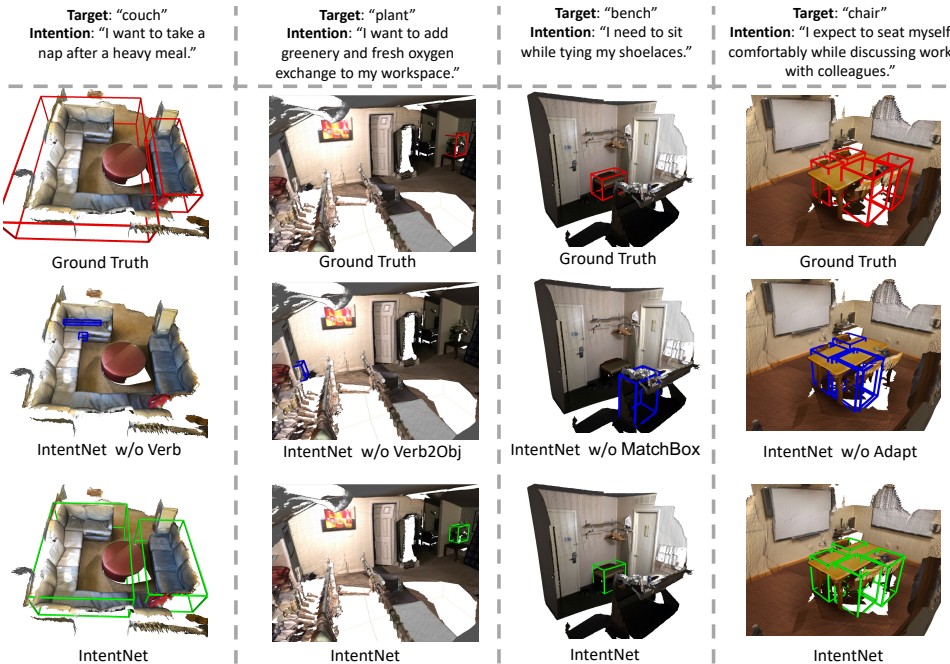

Figure 5: Qualitative results of ablation studies. "Verb" indicates the alignment with verb tokens. "Verb2Obj" indicates the alignment with object token in the sentence when the queries are modulated by the verb. "MatchBox" indicates the Candidate Box Matching, "Adapt" indicates the Cascaded Adaptive Learning. Red boxes indicate the ground truth, Blue boxes indicate the counterpart model's prediction, and Green boxes indicate our model's prediction.

sentence is fundamental for the model to understand the intention, which is learned by $L_{vPos}$ and $L_{vSem}$. Omitting this causes a significant reduction in performance. The comparison between (b) and (e) shows that incorporating reasoning about the object, given the verb, further enhances the model's understanding of the intention. The evaluation between (c) and (e) demonstrates that the Candidate Box Matching module effectively extracts distinctive features near the target boxes to boost overall performance. Lastly, the comparison between (d) and (e) confirms that Cascaded Adaptive Learning facilitates the model to concurrently optimize each of the objectives.

## 5.6 QUALITATIVE RESULTS

We provide qualitative results of ablation studies in Fig. 5 to further demonstrate the effectiveness of each component in our IntentNet. **The first column**: The model without alignment with the verb fails to grasp the intention to identify the target. This highlights the fundamental importance of aligning with verb tokens. **The second column**: The model lacking alignment with object tokens when given verb tokens does not fully capture the intention. In this context, "*add*" is too vague to specifically identify "*plant*". However, with a better understanding of the object "*greenery*", our final model can accurately detect the intended object. **The third column**: The model without the Candidate Box Matching fails to effectively extract distinctive box features near the target, leading it to identify irrelevant items. **The fourth column**: The model lacking Cascaded Adaptive Learning does not efficiently optimize each objective. Hence, despite understanding the intention and reasoning about the target "*chair*" through contrastive learning, it fails to localize all instances of the target due to insufficient optimization of the regression loss.

## 6 CONCLUSION

We propose a new task, 3D Intention Grounding (3D-IG), designed to detect target objects based on human intention within 3D scenes. We introduce a new dataset, Intent3D, with detailed statistics to support this task. To tackle this problem, we first construct comprehensive baselines involving current expert model, foundational model, and LLM-based model. Finally, we propose a novel method, IntentNet, which includes three key aspects: intention comprehension, reasoning to identify object candidates, and cascaded optimization of objective functions, to tackle 3D-IG.

**Acknowledgments:** This research is supported by NSF IIS-2309073, ECCS-2123521 and Cisco Research unrestricted gift. This article solely reflects the opinions and conclusions of its authors and not the funding agencies.

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

# A    APPENDIX

## A.1    ADDITIONAL DISCUSSION OF INTENT3D DATA

**Failed Cases** Here, we aim to provide more details of failed cases in the responses from GPT. As shown in Table 4, (1) GPT sometimes fails to generate a complete sentence. (2) GPT may generate a sentence that includes Unicode characters. (3) GPT may generate a sentence that is not directly related to the object, since the name of the object's category is polysemous. For instance, "*keyboard*" can refer to a musical instrument, but in our dataset, it actually denotes a computer peripheral. (4) We exclude objects whose categories are too general, such as "*machine*", in which case GPT fails to generate a specific intentional text.

Table 4: Examples of failed cases in the responses from GPT.

| | |
|---|---|
| (1) "*bathroom cabinet*": "*I*" | |
| (2) "*chair*" | : "*I need to sit for the duratio\u043d of the conference*" |
| (3) "*keyboard*" | : "*I need to compose music for my upcoming online performance*" |
| (4) "*machine*" | : "*I want to utilize the object (depending on the type) to assist me in tasks*" |

**API Usage** In employing ChatGPT (GPT-4), adjustment of the temperature parameter enables control over output randomness. Higher temperature values foster greater diversity and creativity, while lower values yield more deterministic but potentially repetitive responses. We set temperature=1.2 to optimize the diversity of model outputs. We have explored alternative GPT versions such as gpt-3.5-turbo-1106, and gpt-4-1106-preview. However, empirical evaluation revealed that GPT-4 consistently outperforms others, making it our preferred choice.

**Limitations** The quality and variety of intention sentences generated for each object class depend on the performance of GPT-4. Although GPT-4 demonstrates strong proficiency in real-world knowledge, and we manually refined outputs to address garbled cases, some inherent subjectivity may still be introduced due to GPT-4's training data.

## A.2    ADDITIONAL IMPLEMENTATION DETAILS

Here, we provide additional details on the implementation of each method. For BUTD-DETR [Jain et al., 2022], we adhere to its official configuration in ScanRefer [Chen et al., 2020]. The batch size is set to 24, and the learning rate (which is the same as ours) decreases to one-tenth at the 65th epoch. We follow the suggestion in its source code to continue training the model while monitoring the validation accuracy until the model converges. It finally takes 100 epochs to converge. For EDA [Wu et al., 2023], we also follow its official configuration, where the batch size is set to 48. The learning rate for backbones is 0.002, and for the rest, it is 0.0002. The learning rate decreases to one-tenth at the 50th and 75th epochs. It takes 104 epochs to converge. For 3D-VisTA [Zhu et al., 2023], we use its official configuration in ScanRefer [Chen et al., 2020], where the batch size is set to 64, and the learning rate is 0.0001. The warm-up step is set to 5000. Although the official training epoch is 100, we find that the model converges at the 47th epoch. We adopt its pre-trained checkpoint provided in the source code for fine-tuning. In the case of Chat-3D v2, we follow its official instructions to prepare the annotations. It takes 3 epochs to fine-tune the model.

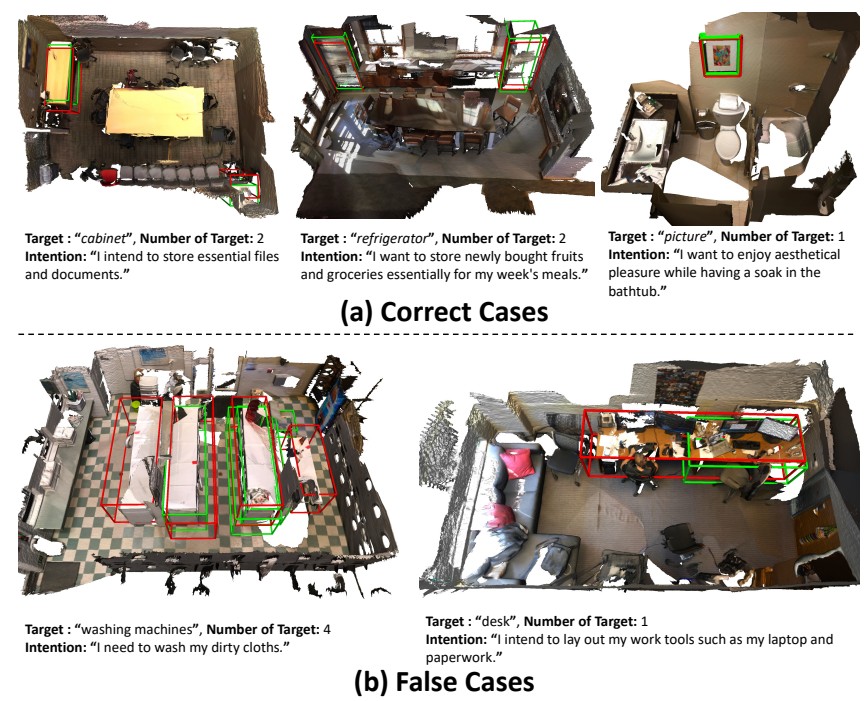

**Target :** *"cabinet"*, **Number of Target:** 2
**Intention: "**I intend to store essential files and documents.**"**

**Target :** *"refrigerator"*, **Number of Target:** 2
**Intention: "**I want to store newly bought fruits and groceries essentially for my week's meals.**"**

**Target :** *"picture"*, **Number of Target:** 1
**Intention: "**I want to enjoy aesthetical pleasure while having a soak in the bathtub.**"**

**(a) Correct Cases**

**Target : "**washing machines**"**, **Number of Target:** 4
**Intention: "**I need to wash my dirty cloths.**"**

**Target : "**desk**"**, **Number of Target:** 1
**Intention: "**I intend to lay out my work tools such as my laptop and paperwork.**"**

**(b) False Cases**

Figure 6: Visualization of correct cases and false cases of IntentNet. Red boxes are ground truth boxes, green boxes are prediction outputs

### A.3 ADDITIONAL QUALITATIVE RESULTS OF INTENTNET

In this section, we present additional qualitative results of our IntentNet, showcasing both correct and false cases.

**Correct Cases** In the upper row (Fig. 6(a)), IntentNet demonstrates impressive accuracy in various scenarios where the number of target instances exceeds one and the instances are not closely situated, as observed in cases of "*cabinet*" or "*refrigerator*". Furthermore, for the less common intention "*enjoy aesthetical pleasure*", IntentNet also performs well.

**False Cases** In the lower row (Fig. 6(b)), we highlight cases where IntentNet fails. Particularly, in the case of "*washing machines*", IntentNet struggles to detect all instances in large environments with multiple targets (four instances). However, we also find that some false cases are attributed to the ambiguity or subjectivity in the bounding box annotations provided by ScanNet [Dai et al., 2017]. For instance, in the "*desk*" case, the annotation includes a double-person desk, while IntentNet detects only one of the single-person desks, which is also reasonable.

### A.4 ADDITIONAL DISCUSSION OF AMBIGUITY

In our dataset, intentions and objects are not one-to-one association. For example, "*I want to sleep*" relates to both "*couch*" in living room and "*bed*" in bedroom. However, the reason we generate intention towards a "*bed*" in a "*bedroom*" that does not have a "*couch*" in it, is to provide a clear and unambiguous mapping of intention to object for better training. As shown in Fig. 7, we input "*I intend to rest my legs*" with the above ambiguous scenario where multiple objects can fulfill this intention. When we lower the confidence score filter to allow multiple outputs, our model can ground many reasonable objects ("*footrest*", "*couch*", "*table*", etc.), since these objects have similar intentions in our training set but just appear in different scenes for unambiguous training.

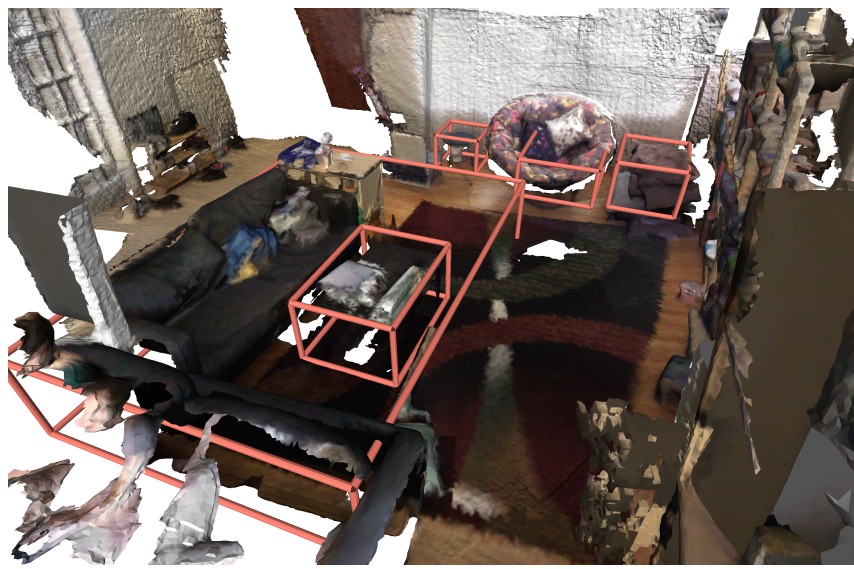

Figure 7: Example for multiple objects prediction.

## A.5 ADDITIONAL DISCUSSION OF HUMAN-LIKENESS

To assess the concern regarding whether ChatGPT-generated sentences exhibit human-like linguistic naturalness, similar to [Qu et al., 2024], we randomly selected 500 original responses produced by ChatGPT and asked three PhD students to manually evaluate them. The results show that all intention sentences meet the criteria for human-like quality.

## A.6 ADDITIONAL COMPARISON WITH GPT-4

Table 5: Compare with two-stage method based on GPT-4.

| Method | Detector | top1-acc@0.25 | top1-acc@0.5 | ap@0.25 | ap@0.5 |
|---|---|---|---|---|---|
| GPT-4 | GroupFree | 41.40 | 28.40 | 15.10 | 7.76 |
| IntentNet | GroupFree | 58.34 | 40.83 | 41.90 | 25.36 |

We provide an additional comparison with a two-stage method based on GPT-4. For the two-stage method, we fairly use the same detector as ours to predict bounding boxes and categories. Given the intention and the detector's predictions, GPT-4 selects the most relevant objects. Results are shown on validation set in Table 5. Due to the detector's prediction errors and hallucinations from GPT-4, it performs poorly in AP metric.

## A.7 CASES WHERE INTENTNET FAILS BUT OTHERS FED WITH OBJECTS SUCCEED

We provide failed cases of our IntentNet to offer potential improvement directions for future work. We find that IntentNet sometimes struggles with intentions involving multiple actions.

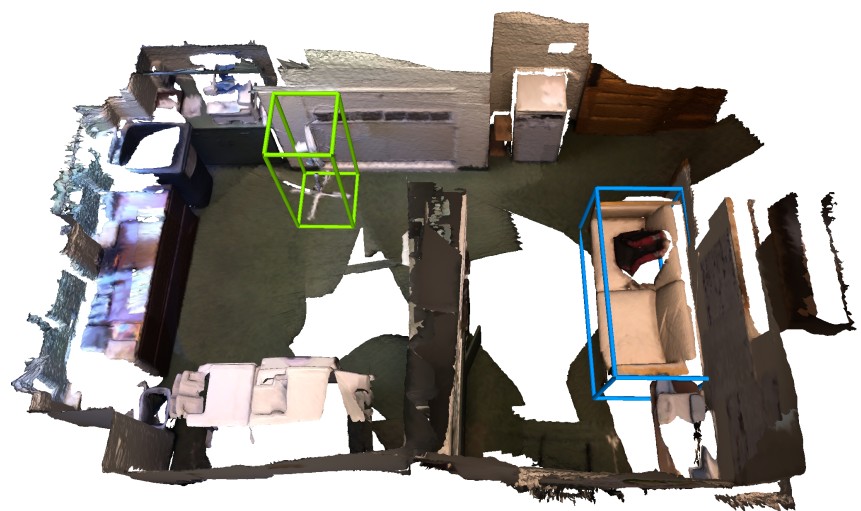

Figure 8: A case where IntentNet fails (blue box) but EDA fed with ground truth object candidates succeeds (green box). The input intention is "I need to cool down on hot days while sitting in the living room."

As shown in Fig. 8, consider the intention: "I need to cool down on hot days while sitting in the living room." In this case, the primary intention is "cool down", which targets the object "fan". However, the secondary action "sitting" acts as a distractor, leading our model to incorrectly detect "couch" (blue bounding box). When we directly provide ground truth object to EDA, the task becomes simplified, allowing it to correctly detect the "fan" (green bounding box). Another example is shown in Fig. 9. The input intention is "I want to sit comfortably while doing my office work." Here, "doing my office work" distracts IntentNet from correctly detecting the optimal object, causing it to wrongly identify "table" instead of "chair", which should be the correct object for "sit comfortably".

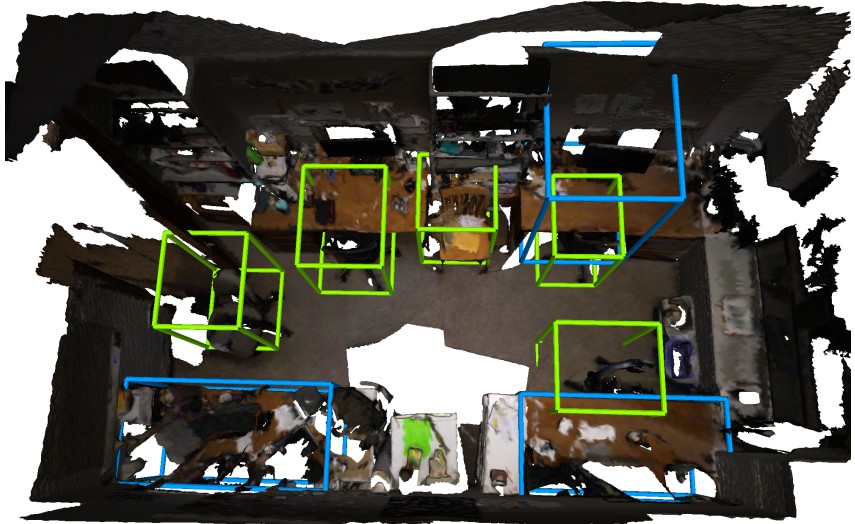

Figure 9: A case where IntentNet fails (blue box) but EDA fed with ground truth object candidates succeeds (green box). The input intention is "I want to sit comfortably while doing my office work."

## A.8    VISUALIZATION OF GROUNDING FOR TRIVIAL OBJECTS

In our data construction, we filter out trivial objects whose number of instances is greater than or equal to six, to increase the dataset's difficulty and quality. However, we want to make it clear that this filtering is independently applied to each scene itself, not to the entire dataset. Specifically, as shown in Fig. 10, in this scene, "table" is a trivial object (with quantity greater than or equal to 6)

and is filtered out during the training phase. But this does not mean that the entire dataset lacks the intention of "table", because in other scenes, "table" may not be numerous, and therefore the intention regarding "table" still exists in the dataset. Thus, in the inference stage, the model is able to detect such common used objects (green boxes). Fig. 11 is another example, in which chairs are trivial objects.

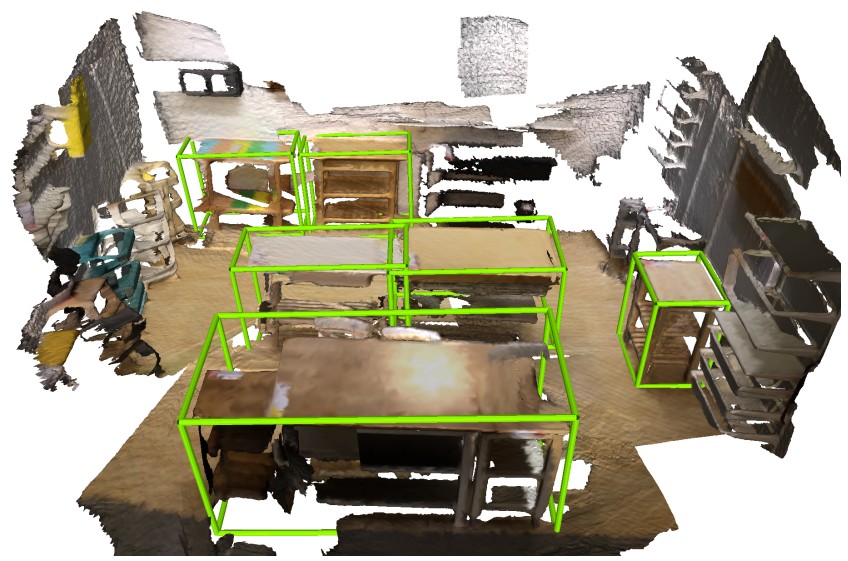

Figure 10: A case where IntentNet grounds trivial objects (green box). The input intention is "I need to have a space where I can spread out and study."

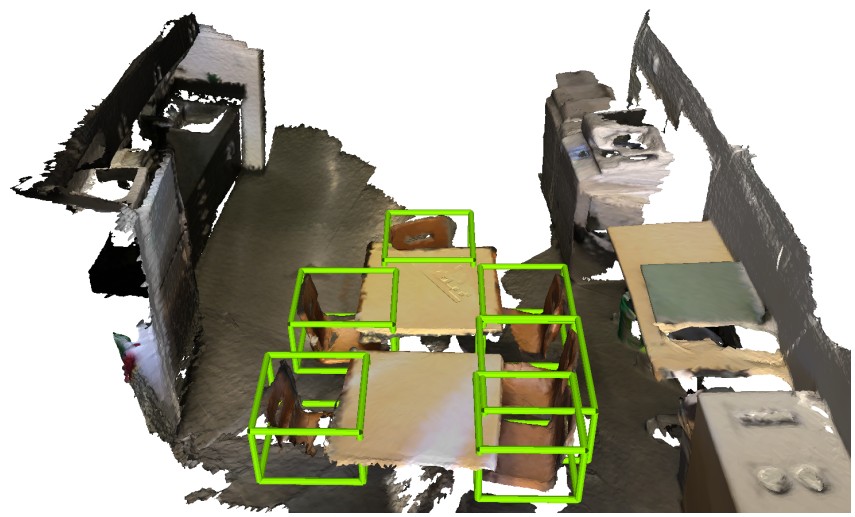

Figure 11: A case where IntentNet grounds trivial objects (green box). The input intention is "I want to sit down during long conferences."

### A.9 DETAILED STATISTICS

We provide detailed statistics of verb and noun toward each object category in Fig. 12 and Fig. 13, respectively. In Fig. 14, we also provide a detailed histogram of different objects we use in constructing our intention sentences. Here, every third data point is displayed for the sake of clarity.

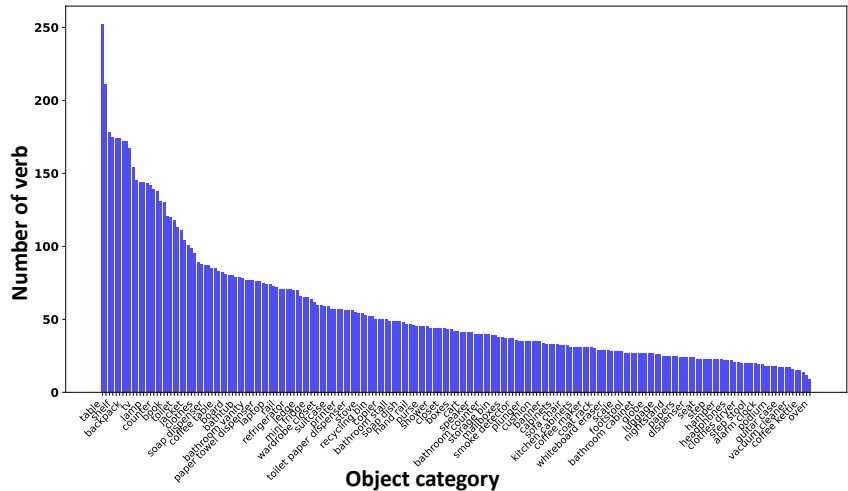

Figure 12: Detailed statistics of verb used in intentions toward each object category

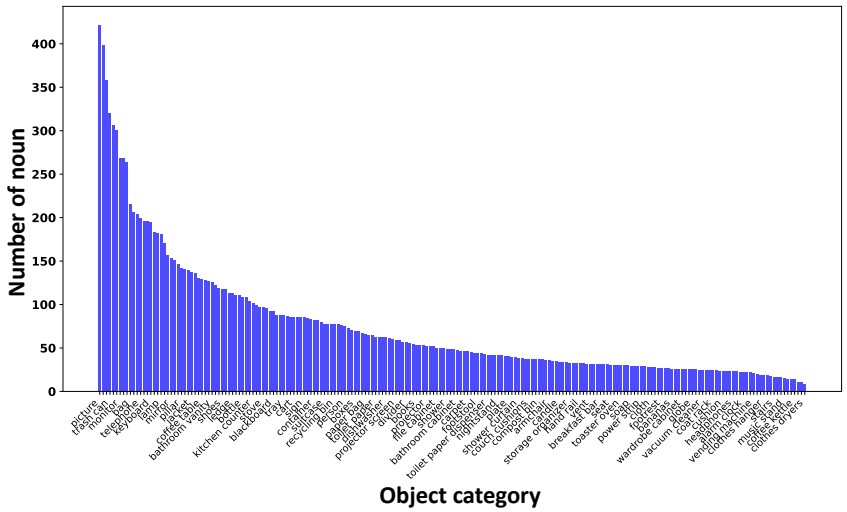

Figure 13: Detailed statistics of noun used in intentions toward each object category

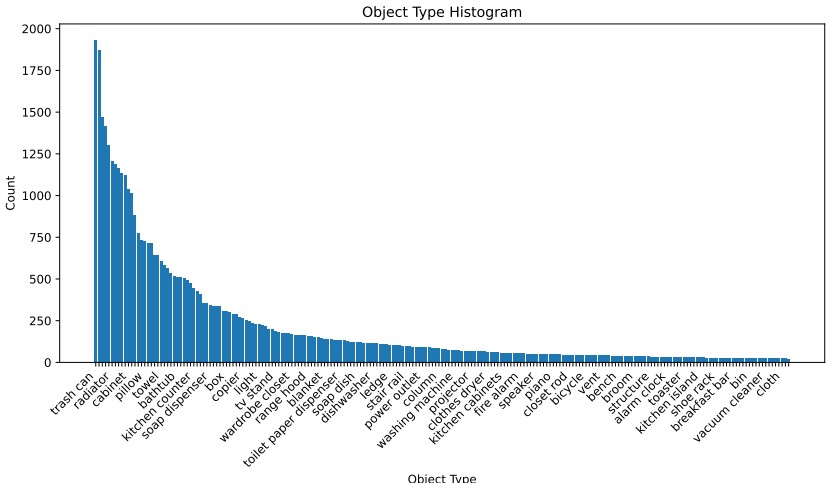

Figure 14: The statistic of object type histogram.

## A.10 MONITOR THE TRAINING PROCESS OF INTENTNET

We provide the accuracy curves and the AP curves of the training process of IntentNet. As shown in Fig. 15 and Fig. 16, the accuracy and AP of IntentNet both converge and achieve strong performances, which indicates that our model trains effectively by the training set.

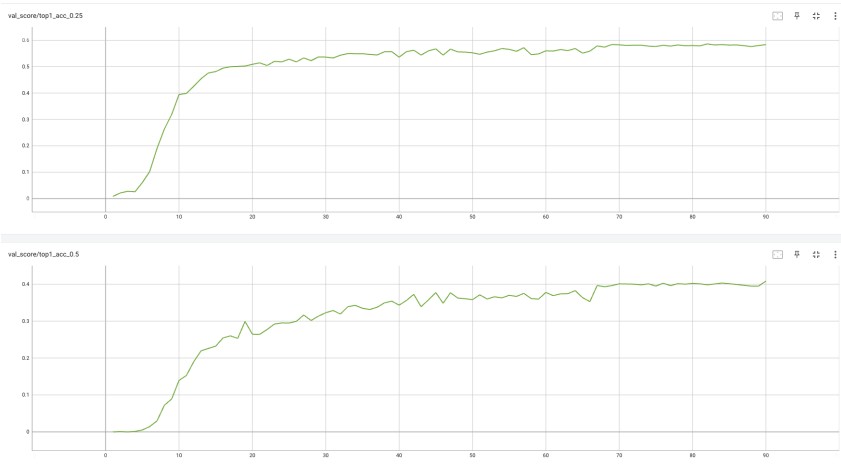

Figure 15: Accuracy curves of the training process of IntentNet.

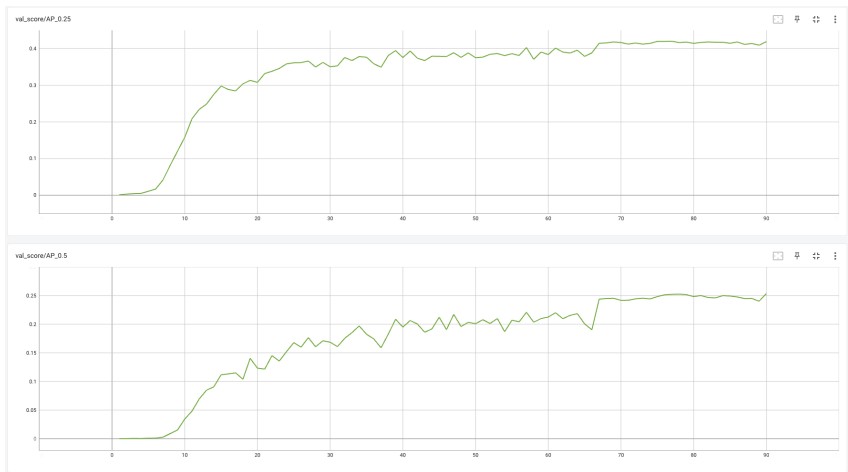

Figure 16: AP curves of the training process of IntentNet.

## A.11 DETAILS OF INTENTNET

As shown in Fig. 17, We provide a detailed figure to illustrate the connections in the network and key modules of our loss functions.

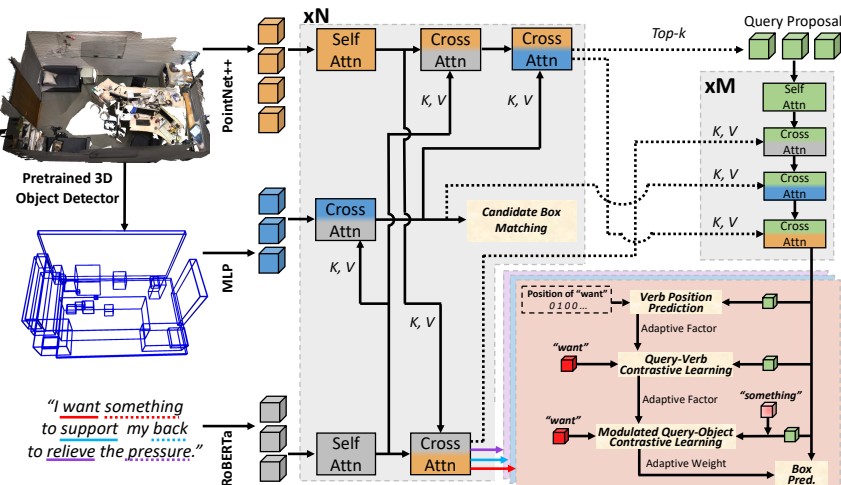

Figure 17: Details of IntentNet. K, V indicate the Key and Value of cross-attention layer, respectively.

