# OpenReview forum: "Intent3D: 3D Object Detection in RGB-D Scans Based on Human Intention"
_ICLR.cc/2025/Conference — ICLR 2025 Poster_

### Official Review · Reviewer_qHNc · 2024-10-20

**Soundness:** 3
**Presentation:** 2
**Contribution:** 3
**Rating:** 6
**Confidence:** 4

**Summary:**

This paper presents a novel framework for 3D object detection that integrates human intention into the detection process. The authors introduce the Intent3D dataset, aiming to enhance the model's understanding of human needs in real-world scenarios. The proposed method, named IntentNet, employs a multi-instance detection approach, where the model is tasked with identifying multiple instances of objects based on free-form textual descriptions of human intentions. The authors evaluate their approach against several baselines, including expert models designed for 3D visual grounding, foundation models for generic 3D understanding tasks, and Large Language Model (LLM)-based models. The evaluation demonstrates the effectiveness of IntentNet in achieving state-of-the-art performance on the Intent3D benchmark.

**Strengths:**

1. The introduction of the 3D Intention Grounding (3D-IG) task could contribute to the 3D visual grounding community.
2. The proposed Intent3D dataset is extensive, comprising 44,990 intention texts linked to 209 fine-grained object classes from 1042 3D scenes. This dataset provides a valuable resource for training and evaluating models in the context of human intention.
3. The proposed modules in IntentNet are generally technically sound.

**Weaknesses:**

1. Object detection based on human intention is indeed a new task. However,  why should we need to have a dataset and a model dedicated to this task? I think the detection based on human intention can be achieved in a modulized manner. For example, first, let the 3D detection module detect all types of objects in the scene. Then ask  LLM to decide the subsets of the detected objects that can fulfill the human intention. I think this could be more flexible compared with training a dedicated detection model based on human intention prompts.
2. In L210, it is mentioned that around six intention texts are generated per object. How do you determine this number (six)? Can it guarantee that all possible intentions can be covered and trained well?
3. The contents around Eq (3) are hard to understand for me. What is t in Eq(3)? Although the authors claimed that the code would be released to facilitate the understanding, it would be better to add a figure to illustrate the connections in the network.
4. Figure 4 is too abstract. Even though many connections and modules are included, it is not very informative. I suggest more detailed diagrams can be provided for the key modules further.
5. Figure 5 shows that the verb alignment is very helpful for the prediction quality. Do you think this is caused by the limited training data? If more data is available for training, would this module still be essential?

**Questions:**

Please refer to the weakness part/

---

> ### Author Response · Authors · 2024-11-21
>
> ## **Response 4.1: Addressing Weakness 1 -- On the Necessity of Intent3D Dataset and IntentNet model dedicated to 3D Intention Grounding**
>
> Thank you for this insightful question. While a modularized approach may seem flexible, it is not well-suited for addressing the fine-grained, multimodal reasoning required in 3D intention grounding. The modularized pipeline, which separates detection and intention reasoning, introduces two critical issues:
>
> 1. Error Accumulation: The pipeline accumulates errors from both the single-modal 3D detector and the LLM's reasoning.
> 2. Hallucination from LLMs: Large language models suffer from the hallucination problem, necessitating rigorous comparisons to evaluate their actual performance.
>
> As demonstrated in Line475, we actually have already conducted experiments with a modularized approach using GPT-4 in Appendix (A.6 page 17). In this setup, a 3D detector identifies all objects in the scene, and GPT-4 determines which objects satisfy the intention. As shown in the table below, the performance of this modularized method is inferior to our IntentNet, underscoring its limitations.
>
> | Model              | Top1-Acc@0.25 | Top1-Acc@0.5 | AP@0.25 | AP@0.5 |
> |--------------------|---------------|--------------|---------|--------|
> | GPT-4 + Proposal   | 41.40         | 28.40        | 15.10   | 7.76   |
> | IntentNet          | 58.34         | 40.83        | 41.90   | 25.36  |
>
> This actually resonates with the trend in visual grounding field:
>
> - In 2D, approaches have evolved from two-stage methods like MAttNet[1] to one-stage methods like SegVG[2], which emphasize necessity of multimodal fusion instead of modularization.
> - Similarly, in 3D, methods have progressed from ScanRefer[3] to EDA[4], showing that advanced approaches increasingly prioritize integrated multimodal reasoning over modularized pipelines.
>
> A dedicated dataset for 3D intention grounding is essential, as it provides a focused benchmark to rigorously drive progress in this multimodal reasoning problem. And our IntentNet provides a strong baseline whose novel contributions in verb reasoning and verb-object reasoning can bring inspirations to our community on tackling this intention grounding problem.
>
> [1] MAttNet: Modular Attention Network for Referring Expression Comprehension, CVPR 2018
>
> [2] SegVG: Transferring Object Bounding Box to Segmentation for Visual Grounding, ECCV 2024
>
> [3] ScanRefer: 3D Object Localization in RGB-D Scans using Natural Language, ECCV 2020
>
> [4] EDA: Explicit Text-Decoupling and Dense Alignment for 3D Visual Grounding, CVPR 2023
>
> ## **Response 4.2: Addressing Weakness 2 -- On the Determination of Six Intentions per Object and the Completeness and Effectiveness of dataset**
>
> In our implementation, we find that six sentences on average are enough to cover diverse intentions toward each object, while preventing repetition. Also, we reference the scale of ScanRefer and Nr3D, two widely used datasets in the field of visual grounding. Based on this scale, an average of 6 intentions per object can result in a dataset of nearly 45K.
>
> As detailed in Section 3.3, our dataset exhibits a rich diversity in sentence structure and verb-noun combinations to show its completeness:
>
> - Verbs: A total of 1,568 distinct verbs are included.
> - Nouns: A total of 2,894 distinct nouns are used.
> - Object category: A total of 209 fine-grained classes.
> - Diversity per category: On average, each object category is associated with 58 unique verbs and 77 unique nouns.
>
> This extensive coverage ensures the dataset covers a wide range of human intentions across different scenarios.
>
> Additionally, the performance in Tables 1 and 2 demonstrates that our model achieves strong results on both validation and test sets, confirming the effectiveness of the generated intentions which are sufficient for training.

---

> ### Author Response · Authors · 2024-11-21
>
> ## **Response 4.3: Addressing Weakness 3 & 4 -- Explain Eq(3) and provide a detailed figure.**
>
> Thank you for your comment regarding Eq. (3). We appreciate the opportunity to clarify it step by step for the meaning of $t$.
> In Eq. (3), $T$ indicates all the text tokens; $T_l$ is the subset of $T$, which indicates the verb tokens among the text tokens. Therefore, the $t$ of “$t \in T$” indicates a single text token which could be a verb or not, and t of “$t \in T_l$” is a single text token which is a verb.
> To provide further clarity, we have included a more detailed figure in the Appendix (A.11, page 22) of the revised paper. We highlight the title of the section in red color for your reference. This figure aims to illustrate the connections in the network and the key modules of our loss functions.
>
> ## **Response 4.4: Addressing Weakness 5 -- On the assumption of scaling effect of Verb Alignment.**
>
> Thank you for raising this insightful question. We think that the effectiveness of the Verb Alignment comes from its dense supervision signal instead of limited data. And we assume it can further bring improvement if we could scale the data in the future.
>
> Firstly, in the 3D domain, acquiring large-scale real-world point cloud data is a significant challenge, as it is not as easily accessible as 2D data from the internet. In this context, exploring breakthroughs to make the best of available labeled data becomes crucial, such as our verb alignment.
>
> Secondly, even if 3D real-world data were to become more abundant in the future, we believe that verb alignment will continue to be effective. This is because verb alignment enhances the utilization of supervision signals from individual sentences.
>
> For traditional detection losses, detecting the target does not necessarily require a comprehensive understanding of the entire intention sentence. For example, in the intention *“I want to sit comfortably while listening to music,”* the target is *“chair.”* The model may only need to focus on the verb *“sit”* to infer the target, but losses from verb alignment encourage a more holistic understanding of the sentence, including the understanding of all verb-object links, e.g. *“listening to music”*. As a result, the supervision signal becomes denser, improving the model's overall reasoning capabilities.

---

> > ### Author Response · Authors · 2024-11-22
> >
> > We appreciate all your suggestions that help us improve our paper. As the deadline for discussion is approaching, we would be glad to address any remaining questions or concerns. Please let us know if there are any further points you'd like to discuss!

---

> ### Comment · Reviewer_qHNc · 2024-11-25
>
> Thanks for the response. The authors addressed my issues. I plan to keep my original rating.

---

> > ### Author Response · Authors · 2024-11-26
> > **Thanks for the acceptance rate!**
> >
> > We appreciate your recognition that our rebuttal addressed your concerns and your decision to maintain the acceptance rate!

---

### Official Review · Reviewer_kRaB · 2024-10-29

**Soundness:** 3
**Presentation:** 3
**Contribution:** 3
**Rating:** 6
**Confidence:** 2

**Summary:**

The paper introduces a 3D Intention Grounding (3D-IG) task and constructs a novel dataset called Intent3D (sourced from ScanNet data and generated using GPT). Additionally, it proposes a baseline model named IntentNet. I am not an expert in 3D-related fields, so my confidence in this review is not very high.

**Strengths:**

1. The overall quality of the paper is high, with clear writing and easy-to-understand presentation.
2. The contribution of the dataset is significant, as it is the first to construct a 3D detection task focused on intention understanding.

**Weaknesses:**

1. More comparisons with recent works should be provided in Tables 1 and 2. Additionally, there is a minor mistake: the detector names “GroupFree” and “Group-Free” in the first two rows of Tables 1 and 2 do not match.
2. The article gives a subtractive ablation experiment. I would like to see an additive ablation experiment, such as how the effect of verb alone works.
3. The article does not give the performance of the proposed IntentNet in traditional 3D grounding.

**Questions:**

1. Will the proposed dataset and code be open-sourced?
2. IntentNet seems to be a two-stage approach, where a pre-trained 3D detector first extracts proposals, which are then matched with text. Are there any one-stage methods that directly fuse 3D data and text to generate boxes? If so, the paper does not seem to provide comparisons with such methods.
3. Regarding the training process, on what hardware was the method trained, and how long did the training take?

---

> ### Author Response · Authors · 2024-11-21
>
> ## **Response 3.1: Addressing Weakness 1 -- Additional Comparisons and revision of typo.**
>
> To provide more comparisons with current LLM-based methods, we compare our IntentNet with LL3DA. Furthermore, we also compare with GPT-4 + Detector Proposal, as another two-stage baseline. Specifically, we use a 3D object detector to proposal objects within the scene. Given the object proposals and the intention, we prompt the GPT-4 to infer the target object. Lastly, we compare with EDA + Detector Proposal, where the object proposals are appended to the intention sentence and input to the baseline, EDA. These results, summarized in the table below, consistently demonstrate the superiority of our IntentNet.
>
> | Model      | val Top1-Acc@0.25 | val Top1-Acc@0.5 | test Top1-Acc@0.25 | test Top1-Acc@0.5 |
> |------------|-------------------|------------------|--------------------|-------------------|
> | LL3DA      | 5.74             | 6.13            | 4.98              | 5.43             |
> | IntentNet  | 58.34            | 40.83           | 58.92             | 42.28            |
>
> | Model              | Top1-Acc@0.25 | Top1-Acc@0.5 | AP@0.25 | AP@0.5 |
> |--------------------|---------------|--------------|---------|--------|
> | EDA + Proposal     | 21.68         | 8.74         | 3.96    | 2.71   |
> | GPT-4 + Proposal   | 41.40         | 28.40        | 15.10   | 7.76   |
> | IntentNet          | 58.34         | 40.83        | 41.90   | 25.36  |
>
> Additionally, we have corrected the typo regarding the detector names (*“GroupFree”*) in Tables 1 and 2 (page 8) in the revised paper. We highlight the revision in red color for your reference. Thank you for pointing this out!
>
> ## **Response 3.2: Addressing Weakness 2 -- Additive Ablation Experiments**
>
> Thank you for the suggestion to include additive ablation experiments. Below, we provide the results of such experiments:
>
> | Baseline | Verb | Verb2Obj | MatchBox | Adapt | Top1-Acc@0.25 | Top1-Acc@0.5 |
> |----------|------|----------|----------|-------|---------------|--------------|
> | ✓        |      |          |          |       | 52.32         | 33.39        |
> | ✓        | ✓    |          |          |       | 54.88         | 35.47        |
> | ✓        | ✓    | ✓        |          |       | 55.98         | 36.10        |
> | ✓        | ✓    | ✓        | ✓        |       | 57.39         | 36.93        |
> | ✓        | ✓    | ✓        | ✓        | ✓     | 58.34         | 40.83        |
>
> Notably, the inclusion of verb alone already provides a significant boost in performance, highlighting its importance in intention grounding.
>
> ## **Response 3.3: Addressing Weakness 3 -- Performance on Traditional 3D Visual Grounding**
>
> Thank you for raising this point. This work primarily focuses on 3D intention grounding, which is an orthogonal task to visual grounding. As such, IntentNet is specifically designed for intention grounding and is not evaluated on traditional visual grounding tasks.
> More concretely, IntentNet focuses on modeling the reasoning between verbs and objects inherent to human intentions, rather than the nouns or adjectives typically emphasized in visual grounding.
> Moreover, we hope that the benchmark contributions and modeling insights presented in this paper can inspire future research on joint training approaches for both intention grounding and visual grounding tasks.
>
> ## **Response 3.4: Addressing Question 1 -- Open Source**
>
> Yes, we will open source our dataset and code. Thank you for your interest!
>
> ## **Response 3.5: Addressing Question 2 -- Clarifying One-Stage Methods.**
>
> Thank you for pointing this out. However, this seems to be a misunderstanding.
>
> [Our method] IntentNet is a one-stage method, as bounding box predictions are generated in an end-to-end manner rather than relying on a pre-trained detector for inference.
> As stated in Line 309 of the paper, the pre-trained detector is only used to provide additional visual references, and the box matching with text is implemented as an auxiliary loss, which does not participate in the actual inference process.
>
> [Other one-stage methods] Moreover, we have already compared IntentNet with existing one-stage methods, such as BUTD and EDA. These models also directly fuse 3D data and text to predict bounding boxes, using a pre-trained detector in a similar way to just provide visual references.
>
> ## **Response 3.6: Addressing Question 3 -- Training Hardware and Time Cost**
>
> Thank you for your question. The training of IntentNet was conducted on 4 NVIDIA A6000 GPUs. The training process takes approximately 24 hours.

---

> > ### Author Response · Authors · 2024-11-22
> >
> > We appreciate all your suggestions that help us improve our paper. As the deadline for discussion is approaching, we would be glad to address any remaining questions or concerns. Please let us know if there are any further points you'd like to discuss!

---

> > > ### Author Response · Authors · 2024-11-25
> > > **Only One Day Remaining, Please Take a Look at our Rebuttal!**
> > >
> > > Dear Reviewer,
> > >
> > > Thank you for dedicating your time reviewing our paper. As the discussion period deadline is approaching, we kindly invite any further comments or concerns you might have. Your feedback has been immensely valuable to us in refining the paper.
> > >
> > > Best,
> > >
> > > The Authors

---

### Official Review · Reviewer_tzNR · 2024-10-31

**Soundness:** 2
**Presentation:** 2
**Contribution:** 2
**Rating:** 5
**Confidence:** 2

**Summary:**

This paper introduces the task of 3D Intention Grounding (3D-IG), which aims to automate the reasoning and detection of target objects in real-world 3D scenes using human intention cues. To this end, the authors constructed the Intent3D dataset, comprising 44,990 intention texts across 209 fine-grained object categories, and developed several baseline models to evaluate various 3D object detection techniques. Finally, the authors proposed a novel method, IntentNet, which optimizes intention understanding and detection tasks through techniques such as verb-object alignment and candidate box matching, achieving state-of-the-art performance on the Intent3D benchmark.

**Strengths:**

1: A new task in 3D object detection employing RGB-D, based on human intention, facilitates smoother and more natural communication between humans and intelligent agents.

2:The author propose a high-quality vision-language dataset and focuses on the human’s intention for 3D object detection, which will facilitate the progress of 3D scene understanding.

**Weaknesses:**

1: There has been a few methods to combine 3D scene understanding with LLM beyond Chat3D v2, such as LL3DA, Grounded 3D-LLM, ReGround3D and so on.  The paper does not highlight the advantages compared to them.

2: The object selection method is too crude, as it removes some commonly used objects by humans when filtering Non-trivial Objects. Figures 3 (d) and (e) indicate that the dataset lacks sufficient diversity in the types of objects included.

3: The limited variety of object category included in the dataset, fails to demonstrate the grounding effect for the missing objects in the dataset.

**Questions:**

1：There has been a few methods to combine 3D scene understanding with LLM beyond Chat3D v2, such as LL3DA, Grounded 3D-LLM, ReGround3D and so on.  What are the advantages of this paper compared to theirs? More experimental evidence is needed to demonstrate the advantages of the this paper over other methods.

2: Current large language models can also directly infer human intentions; what are the advantages of this paper compared to them?

3：When filtering Non-trivial Objects, objects with more than six instances in fine-grained categories are directly removed, which may lead to the exclusion of commonly used objects. Could we consider adding more fine-grained descriptions for these objects instead of outright deletion?

4:  Figures 3 (d) and (e) indicate that the dataset lacks sufficient diversity in the types of objects included. Experimental validation is necessary to determine whether the variety of object types included in the dataset is sufficient for the model to learn effectively.

---

> ### Author Response · Authors · 2024-11-21
>
> ## **Response 2.1: Addressing Weakness 1 and Question 1 -- Advantages Compared to LLM-Based Multimodal Methods**
>
> [Advantages]
> LLM-based multimodal methods, such as Chat3D-v2 or the GPT-4 two-stage approach (Line475), primarily operate at the object level. This dependency on off-the-shelf detectors for object proposals introduces detector’s error and also limits their fine-grained multimodal fusion. Lack of multimodal fusion leads to their poor performance. Such kind of shortage have also been witnessed in 2D domian. For example, previous works including POPE[1] reveal that multimodal LLMs often hallucinate in the basic object existence problem due the lack of fusion. This challenge becomes more pronounced in fine-grained multimodal tasks like our 3D intention grounding, where comprehensive feature fusion and reasoning are critical.
> In contrast, IntentNet addresses this gap through its explicit intention modeling and fine-grained feature integration. Specifically, IntentNet employs point-word-level fusion and enables the model to reason over verb-object relationships explicitly.
>
> [Experimental Evidence]
> Due to resource constraints during the rebuttal period, we choose to conduct a comparison with LL3DA. The results, summarized in the table below, show that IntentNet significantly outperforms LL3DA. Although LL3DA uses point-level features, its Interactor3D module relies on an off-the-shelf detector during inference for bounding box proposals. Errors from this detector propagate to the final predictions, limiting its effectiveness.
>
> | Model      | val Top1-Acc@0.25 | val Top1-Acc@0.5 | test Top1-Acc@0.25 | test Top1-Acc@0.5 |
> |------------|-------------------|------------------|--------------------|-------------------|
> | LL3DA      | 5.74             | 6.13            | 4.98              | 5.43             |
> | IntentNet  | 58.34            | 40.83           | 58.92             | 42.28            |
>
> [Clarification on ReGround3D]
> Thank you for highlighting ReGround3D. As it is published within four months of our ICLR submission, it qualifies as a contemporaneous work under the ICLR 2025 policy, and we are not required to compare with it in our submission. Nevertheless, we will include its citation in the camera-ready version.
>
> [1] Evaluating Object Hallucination in Large Vision-Language Models, EMNLP 2023
>
> ## **Response 2.2: Addressing Question 2 -- Advantages Compared to Large Language Models**
>
> While LLMs exhibit strong reasoning capabilities for pure language tasks, 3D intention grounding is a multimodal, fine-grained problem that requires joint reasoning across both vision and language. Relying solely on single-modality, language, for reasoning is insufficient.
> For example, consider the intention: *"I want to sit down and rest."* In a *"living room"*, the target object might be a *"sofa"*, while in a *"bedroom"*, it could be a *"chair"* or *"bed"*. Without fusing multimodal information, such as scene context, language-only LLMs cannot accurately ground intentions in 3D space.
> In contrast, our IntentNet explicitly integrates fine-grained point-level features with text features, enabling verb-object pair reasoning to analyze intentions. This joint feature fusion and reasoning framework allows IntentNet to outperform LLMs in this multimodal fine-grained scenarios.

---

> > ### Author Response · Authors · 2024-11-21
> >
> > ## **Response 2.3: Addressing Weakness 2&3 and Question 3&4 -- Object Selection and Dataset Diversity**
> >
> > We believe there might be some misunderstanding regarding the *non-trivial objects* and *object diversity* in our dataset.
> >
> > [Non-Trivial Objects]
> > As stated in Line 187, non-trivial objects are filtered on a per-scene basis. This means an object considered trivial in one scene might not be trivial in another. For example, *"chairs"* are commonly used objects. In a *"conference room"* scene, where chairs are abundant, they are deemed trivial and excluded to improve dataset quality and difficulty. However, in a *"bedroom"* scene, chairs are not trivial and are retained in the dataset. Therefore, our data does have intentions toward commonly used objects.
> > To further clarify, we have added visualizations of IntentNet’s grounding on trivial objects in Appendix A.8 (page 19), with the section title highlighted in red for your reference. These examples demonstrate that our model can accurately ground trivial objects when required.
> >
> > [Object Diversity]
> > Regarding the diversity of objects, *Figure 3 (d) and (e)* do not represent all object categories in our dataset. As indicated in Line 253, the x-axis shows *“every tenth data point”* for clarity. Our dataset actually includes *209 fine-grained classes*, as mentioned in Line 241.
> > For additional details, we provide a detailed version of *Figure 3 (d) and (e)* and an additional histogram of object category in Appendix A.9 (page 20), with the section title also highlighted in red for your reference. These supplementary statistics demonstrate the richness and diversity of our dataset across object categories.
> > It is worth noting that, in contrast, Referit3D is a commonly used 3D Visual Grounding dataset, and they only have 76 types.
> >
> > [Experimental Validation]
> > As shown in Table 1&2, we have constructed a validation set and a test set to benchmark our 3D intention grounding, in which our IntentNet serves as a strong baseline for current benchmark. Specifically, IntentNet achieves 58.34% Top1-Acc@0.25 in the validation set and 58.92% Top1-Acc@0.25 in the test set, which provides sufficient experimental validation to show that our model has learned effectively.
> > In Appendix A.10 (page 21) with the section title highlighted in red for your reference, we further provide the Accuracy curve and AP curve of training process of IntentNet, which finally converge and achieve strong performances, indicating that our model has well trained by the training set.
> >
> > ## **Response 2.4: Addressing Question 3 -- Consideration of Fine-Grained Descriptions for trivial Objects**
> >
> > Thank you for this intriguing suggestion! Introducing fine-grained intentions for those trivial objects could indeed be a viable approach, if the intentions are sufficiently specific to make them unambiguous.
> > Here we provide a feasible example:
> > Consider a large living room containing numerous tables. By calculating the number of points of each table, we could select the largest table. For an intention such as *“I want to find the most comfortable place to put all the food for sharing with friends”*, this *"table"* would no longer be considered trivial, as its larger size makes it distinct from other tables and suitable for this intention.
> > We appreciate this perspective and will incorporate such considerations into the next version of our dataset to further enhance its complexity. However, as highlighted in *Response 2.3*, our current dataset already includes a diverse and extensive range of objects, ensuring comprehensive coverage for this task.

---

> > > ### Author Response · Authors · 2024-11-22
> > >
> > > We appreciate all your suggestions that help us improve our paper. As the deadline for discussion is approaching, we would be glad to address any remaining questions or concerns. Please let us know if there are any further points you'd like to discuss!

---

> > > > ### Author Response · Authors · 2024-11-25
> > > > **Only One Day Remaining, Please Take a Look at our Rebuttal!**
> > > >
> > > > Dear Reviewer,
> > > >
> > > > Thank you for dedicating your time reviewing our paper. As the discussion period deadline is approaching, we kindly invite any further comments or concerns you might have. Your feedback has been immensely valuable to us in refining the paper.
> > > >
> > > > Best,
> > > >
> > > > The Authors

---

> > > > > ### Author Response · Authors · 2024-11-27
> > > > > **[Urgent] ICLR Extends Rebuttal Period – We Are Waiting For Your Response!**
> > > > >
> > > > > Dear Reviewer,
> > > > >
> > > > > ICLR has extended the rebuttal period to encourage your response. We sincerely appreciate the time and effort you have already invested, and we have actively addressed all your concerns in detail.
> > > > >
> > > > > May we ask if you have any remaining questions or issues for clarification? Notably, **Reviewer kRaB** acknowledged that **most of their issues have been addressed** and decide to **raise the score to an acceptance rate**. Similarly, **Reviewer qHNc** has expressed that **their concerns have been resolved** and decide to **maintain the acceptance rate**.
> > > > >
> > > > > Your response is critical for determining the outcome of our paper, and we value your input greatly!
> > > > >
> > > > > Best,
> > > > >
> > > > > The Authors

---

> > > > > > ### Author Response · Authors · 2024-12-02
> > > > > > **[ICLR 2025] Only Two Days Left in ICLR Extended Discussion Period, Please Take a Look at the Rebuttal!**
> > > > > >
> > > > > > Dear Reviewer,
> > > > > >
> > > > > > **We have addressed all your concerns**, including comparison with other methods and clarification of dataset construction. **As your confidence score is low**, we kindly encourage you to review our rebuttal carefully and consider making a more deliberate decision based on our rebuttal.
> > > > > >
> > > > > > *(Ignoring our rebuttal not only diminishes the purpose of the discussion period but also undermines the collaborative spirit of our ICLR community. Additionally, it discredits the sustained effort we have invested over the past three weeks in addressing all your concerns.)*
> > > > > >
> > > > > > **PLEASE**, we sincerely hope to hear from you soon and appreciate your engagement in this important step.

---

> ### Author Response · Authors · 2024-12-02
> **[ICLR 2025] Last Day in ICLR Extended Discussion Period, Please Adjust Your Rate!**
>
> Dear Reviewer,
>
> Today is the final day of the ICLR rebuttal period. We noticed that your confidence score is relatively low, and we sincerely hope you might consider aligning with **Reviewer qHNc**, who has higher confidence and set their score as 6.
>
> **Please adjust your rate** since the authors have completely responded your concerns and you have no more follow-up questions.
>
> For your convenience, we have summarized our contributions and rebuttal in the comment: **Summary of Our Contributions and Rebuttal Responses**.
>
> Best regards,
>
> The Authors

---

### Official Review · Reviewer_gGZj · 2024-11-04

**Soundness:** 3
**Presentation:** 3
**Contribution:** 3
**Rating:** 5
**Confidence:** 3

**Summary:**

This paper introduces a new task named 3D-intention grounding, which is 3D object-detection from direct human-intentions. The paper collects Intent3D dataset which includes 1042 scenes from ScanNet and corresponding paired human-intentions questions and 3D object detections answers. IntentNet is proposed to tackle 3D-intention grounding task by candidate box-matching, verb-object alignment and cascaded adaptively learning.

**Strengths:**

1. The paper presents a clear motivation for 3D-intention grounding, and it includes clear illustrations and presentations of dataset collection procedure.

2. Soundness of each component design of the IntentNet, and thoroughly ablations on each component of the proposed pipeline design.

3. Extensive experiments and discussions demonstrate the effectiveness of the proposed framework compared to different types of baselines.

**Weaknesses:**

Major concern:

I am concerned about possible baseline unfair comparison in the experiment section. Most baselines are designed to tackle nouns-types of questions instead of human-intention types of questions. What if we pass the question to a finetuned LLM and let it infers what types of nouns/objects the question is targeting at from possible objects in a scene detected by existed 3D object detectors? The possible performance of these baselines might be much higher after it is given the object it is expected to detect in a scene.

Minor concern:
It would be interesting if the author can provide some cases where IntentNet fails but other models succeed. Particularly if other models are fed with object/noun directly.

**Questions:**

Please see the weakness section.

I am willing to increase my score if the author resolves my major concern with some additional baseline experimentations.

---

> ### Author Response · Authors · 2024-11-21
>
> ## **Response 1.1: Addressing Major Concern -- Comparison with LLM + Detector**
>
> Thank you for raising this thoughtful concern regarding the potential baseline involving LLM + Detector and the fairness of our comparisons.
>
> [Baseline] To address this, we want to highlight that in Line475, we mention that we have already conducted the LLM + Detector baseline, as detailed in Appendix (A.6 page 17). Specifically, we use a detector to propose objects in the scene. Given the proposed objects and the intention, we prompt one of the most powerful LLMs, GPT-4, to infer which types of objects the question targets. As shown in Table 5 (page 17) or the table below, this two-stage method performs worse than our approach due to detector errors and hallucinations from GPT-4.
>
> | Model              | Top1-Acc@0.25 | Top1-Acc@0.5 | AP@0.25 | AP@0.5 |
> |--------------------|---------------|--------------|---------|--------|
> | GPT-4 + Proposal   | 41.40         | 28.40        | 15.10   | 7.76   |
> | IntentNet          | 58.34         | 40.83        | 41.90   | 25.36  |
>
> [Fairness] Furthermore, we would like to clarify that 3D-VisTA and Chat3D-v2 in our comparisons are general-purpose models designed to handle general language, not specifically for noun-based language. As such, we believe that our comparisons with these models are fair.
>
>
> ## **Response 1.2: Addressing Major Concern -- Comparison with baseline + provided objects**
>
> To address the concern of whether the baseline might be improved given the objects expected to detect in a scene, we conducted an additional experiment, indicated as “EDA + Proposal”. Here, nouns/objects proposed by a detector are appended to the intention sentence as the input for EDA. The results are as below:
>
> | Model            | Top1-Acc@0.25 | Top1-Acc@0.5 | AP@0.25 | AP@0.5 |
> |------------------|---------------|--------------|---------|--------|
> | EDA + Proposal   | 21.68         | 8.74         | 3.96    | 2.71   |
> | EDA              | 43.11         | 18.91        | 14.02   | 5.00   |
> | IntentNet        | 58.34         | 40.83        | 41.90   | 25.36  |
>
> As shown in EDA + Proposal, directly providing nouns/objects expected to detect to EDA does not improve its performance, but rather introduces noise due to detector errors, which interferes model's reasoning. Actually, existing baselines (e.g., EDA, BUTD-DETR) have already use detectors to obtain "objects expected to detect" indirectly.
>
> ## **Response 1.3: Addressing Minor Concern -- Cases where IntentNet fails but others fed with objects succeed**
>
> Thank you for this suggestion. We actually have already provided some failed cases of our IntentNet in the Appendix (Section A.2, Fig. 6, page 16). In our revised paper, we further include visualizations of failed cases in Figure 8 (page 18). We highlight the title of the section in red color for your reference.
> Specifically, we observed that our model sometimes struggles with intentions involving multiple actions. For example, consider the intention:
> *"I need to cool down on hot days while sitting in the living room."*
> In this case, the primary intention is *"cool down"*, which targets the object *"fan"*. However, the secondary action *"sitting"* acts as a distractor, leading our model to incorrectly detect *"couch"* (blue bounding box).
> When we directly provide ground truth object noun to EDA, the task becomes simplified, allowing it to correctly detect the *"fan"* (green bounding box).
> We have included additional visualized examples in Figure 9 (page 18) for further reference.

---

> > ### Author Response · Authors · 2024-11-22
> >
> > We appreciate all your suggestions that help us improve our paper. As the deadline for discussion is approaching, we would be glad to address any remaining questions or concerns. Please let us know if there are any further points you'd like to discuss!

---

> > > ### Author Response · Authors · 2024-11-25
> > > **Only One Day Remaining, Please Take a Look at our Rebuttal!**
> > >
> > > Dear Reviewer,
> > >
> > > Thank you for dedicating your time reviewing our paper. As the discussion period deadline is approaching, we kindly invite any further comments or concerns you might have. Your feedback has been immensely valuable to us in refining the paper.
> > >
> > > Best,
> > >
> > > The Authors

---

> ### Author Response · Authors · 2024-11-27
> **[Urgent] ICLR Extends Rebuttal Period – We Are Waiting For Your Response!**
>
> Dear Reviewer,
>
> ICLR has extended the rebuttal period to encourage your response. We sincerely appreciate the time and effort you have already invested, and we have actively addressed all your concerns in detail.
>
> Since **you have expressed the willingness to raise the score** and we have already completely replied, is there any remaining problem you want to ask? Notably, **Reviewer kRaB** acknowledged that **most of their issues have been addressed** and decide to **raise the score to an acceptance rate**. Similarly, **Reviewer qHNc** has expressed that **their concerns have been resolved** and decide to **maintain the acceptance rate**.
>
> Your response is critical for determining the outcome of our paper, and we value your input greatly!
>
> Best,
>
> The Authors

---

> > ### Author Response · Authors · 2024-12-02
> > **[ICLR 2025] Only Two Days Left in ICLR Extended Discussion Period, Please Take a Look at the Rebuttal!**
> >
> > Dear Reviewer,
> >
> > **We have addressed all your concerns**, including comparison with baselines and failed cases analysis. As **you previously indicated a willingness to raise your score**, we kindly ask if there is anything further you want us to clarify to support this adjustment.
> >
> > *(Ignoring our rebuttal not only diminishes the purpose of the discussion period but also undermines the collaborative spirit of our ICLR community. Additionally, it discredits the sustained effort we have invested over the past three weeks in addressing all your concerns.)*
> >
> > **PLEASE**, we sincerely hope to hear from you soon and appreciate your engagement in this important step.

---

> ### Author Response · Authors · 2024-12-02
> **[ICLR 2025] Last Day in ICLR Extended Discussion Period, Please Adjust Your Rate!**
>
> Dear Reviewer,
>
> Today is the last day of ICLR rebuttal. Given that you previously indicated a willingness to raise your score, **please adjust your rate** since the authors have completely responded your concerns and you have no more follow-up questions.
>
> Best regards,
>
> The Authors

---

### Author Response · Authors · 2024-11-24
**Summary of Our Contributions and Rebuttal Responses**

Dear PCs, ACs, and Reviewers,

We have submitted detailed responses during the rebuttal stage.
**We appreciate the recognition from Reviewer qHNc & kRaB, *currently the only two responsive reviewers*, for acknowledging that our rebuttal addressed their concerns and for their decision on maintain/raise to the acceptance rate.**
Unfortunately, we have not yet received the other two reviewers' feedback. It is worth noting that **[** Reviewer gGZj **]** **indicated a willingness to raise their score based on the rebuttal**. We also invested significant resources, including renting numerous servers at considerable expense, to conduct the experiments requested by the reviewers. It is disappointing that we have not received the rest reviewers' responses. Your attention to this matter is greatly appreciated.

To facilitate further discussion among PCs, ACs, and reviewers, and to assist in finalizing the decision regarding the acceptance of our paper, we would like to briefly recapitulate our contributions and highlight key responses to the questions raised by **[** Reviewer gGZj (R1), Reviewer tzNR (R2), Reviewer kRaB (R3), and Reviewer qHNc (R4) **]**:

### **Summary:**

**(Contribution)**

- In this paper, we propose a new task, 3D Intention Grounding (3D-IG), for detecting objects in 3D real-world environments based on human intention instructions. We greatly appreciate **[** R1 (Strengths 1), R2 (Strengths 1), and R4 (Strengths 1) **]** recognize our **clear motivation** and its **contribution to our community**.
- To systematically study this new problem, we develop the Intent3D dataset to support both the training and benchmarking of 3D-IG. We are thankful that **[** R1 (Strengths 1) **]** acknowledge our dataset collection procedure, **[** R2 (Strengths 2) **]** describe our **dataset as high-quality**, **[** R3 (Strengths 2) **]** emphasize the **dataset’s significant contribution**, and **[** R4 (Strengths 2) **]** note that the **dataset is extensive** and **provides a valuable resource** for this task.
- After building the benchmark, we conduct a comprehensive evaluation of the existing methods. We appreciate **[** R1 (Strengths 2) **]** for recognizing our discussions of **extensive baselines**.

**(Soundness)**

- We propose IntentNet, a strong baseline that achieves SOTA on Intent3D. IntentNet introduces three novel modules: Candidate Box Matching, Verb-Object Alignment, and Cascaded Adaptive Learning. We are grateful for the recognition of IntentNet’s **soundness and effectiveness** from **[** R1 (Strengths 2), R4 (Strengths 3) **]**.
- Finally, to validate the effectiveness of each module in IntentNet, we conducted extensive ablation studies. We thank **[** R1 (Strengths 2 & 3) **]** for acknowledging our **thorough and extensive ablation study** as well as the **effectiveness** of each module.

**(Presentation)**

- We appreciate **[** R1 (Strengths 1) **]** for recognizing our **clear presentation** and **[** R3 (Strengths 1) **]** for acknowledging our **paper's overall high quality** with **clear writing** and **ease of understanding**.

### **Rebuttal:**

**(New experiments or results)**

- Added experiments comparing with LLM + detector [Response 1.1, 3.1, 4.1].
- Added experiments comparing with baseline + detector [Response 1.2, 3.1].
- Added experiments comparing with LLM-based methods [Response 2.1, 3.1].
- Conducted additional ablation studies [Response 3.2].
- Included analysis of failed cases [Response 1.3].
- Provided detailed demonstrations [Response 4.3].

**(Clarify misunderstandings)**

- Clarified our data selection [Response 2.3].
- Clarified our data diversity [Response 2.3].
- Clarified the comparison with one-stage methods [Response 3.5].

**(Explain questions)**

- Explained our advantages over other methods [Responses 2.1, 2.2].
- Explained the fairness of comparisons [Responses 1.1, 2.1].
- Explained the effectiveness of our dataset [Responses 2.3, 4.2].
- Explained potential improvements [Response 2.4].
- Explained validation on visual grounding tasks [Response 3.3].
- Explained open-source plans and experimental details [Responses 3.4, 3.6].
- Explained the design of dataset construction [Response 4.2].
- Explained the scaling problem [Response 4.4].

---

### Author Response · Authors · 2024-11-28
**[Urgent] Please encourage the reviewers to participate in the rebuttal discussion.**

Dear Area Chair,

As the extended Discussion Period is nearing its end, I kindly request your assistance in prompting **[** Reviewer gGZj **]** and **[** Reviewer tzNR **]** to respond to my comments.

Notably,
- **The only two responseive reviewers**, **[** Reviewer kRaB **]** and **[** Reviewer qHNc **]**, **decide to accept** my paper after reviewing my rebuttal. **[** Reviewer qHNc **]** has **the highest confidence score** toward our paper.
- **[** Reviewer gGZj **]** had previously indicated **a willingness to raise their score** based on my rebuttal and has **the highest confidence rate** among the reviewers who gave a score of 5. But they remain unresponsive.
- **[** Reviewer tzNR **]** remains unresponsive and has the lowest confidence rate.

Despite my repeated attempts to engage **[** Reviewer gGZj **]** and **[** Reviewer tzNR **]**, I have not yet received any response. Your help in expediting their feedback would be greatly appreciated.

Thank you for your time and support.

Best regards,

---

### Meta-Review · Area_Chair_iohH · 2024-12-20

**Metareview:**

This paper introduces the 3D Intention Grounding (3D-IG) task, which aims to detect objects in 3D scenes based on human intention instructions. To support this task, the authors develop the Intent3D dataset and propose a method called IntentNet, which consists of three modules: candidate box matching, verb-object alignment, and cascaded adaptive learning.

Initial reviewer concerns focused on several aspects of the paper, including:

-	Unfair baseline comparisons (gGZj)
-	The motivation behind the 3D-IG task and the Intent3D dataset (gGZj)
-	The contribution of the verb-object alignment module (qHNc)
-	Clarifications regarding the advantages over large language models (LLMs) (tzNR)
-	Object selection and dataset diversity (tzNR)
-	The training process (kRaB)
-	Issues with equations and figures (qHNc)
-	Request for failure case analysis (gGZj)
-	The need for additional baseline comparisons (tzNR, kRaB)
-	The request for further ablation studies (kRaB)
-	Evaluation on traditional visual grounding tasks (tzNR)

The authors addressed these concerns comprehensively in their rebuttal, providing clarifications and additional experimental results where requested. Reviewer qHNc acknowledged that their concerns had been addressed, while the other three reviewers (gGZj, tzNR, and kRaB) did not respond to the rebuttal. After carefully reviewing the authors' responses and the remaining concerns, the AC believes that most issues have been adequately addressed.

Given that Intent3D represents a new and promising task for embodied AI, along with its contribution of a novel dataset and a strong baseline method, the AC recommends accepting this paper.

**Additional Comments On Reviewer Discussion:**

Initial reviewer concerns focused on several aspects of the paper, including:

1. Unfair baseline comparisons (gGZj)
2. The motivation behind the 3D-IG task and the Intent3D dataset (gGZj)
3. The contribution of the verb-object alignment module (qHNc)
4. Clarifications regarding the advantages over large language models (LLMs) (tzNR)
5. Object selection and dataset diversity (tzNR)
6. The training process (kRaB)
7. Issues with equations and figures (qHNc)
8. Request for failure case analysis (gGZj)
9. The need for additional baseline comparisons (tzNR, kRaB)
10. The request for further ablation studies (kRaB)
11. Evaluation on traditional visual grounding tasks (tzNR)

The authors addressed these concerns comprehensively in their rebuttal, providing clarifications and additional experimental results where requested. Reviewer qHNc acknowledged that their concerns had been addressed, while the other three reviewers (gGZj, tzNR, and kRaB) did not respond to the rebuttal. After carefully reviewing the authors' responses and the remaining concerns (points 1,2,4-6,8-11), the AC believes that most issues have been adequately addressed.

Considering Intent3D represents a new and promising task for embodied AI, along with its contribution of a novel dataset and a strong baseline method (agreed by most reviewers), the AC recommends accepting this paper.

---

### Decision · Program_Chairs · 2025-01-22

Accept (Poster)